# LOW-COST HIGH-POWER MEMBERSHIP INFERENCE BY BOOSTING RELATIVITY

## ABSTRACT

We present a robust membership inference attack (RMIA) that amplifies the distinction between population data and the training data on any target model, by effectively leveraging both reference models and reference data in our likelihood ratio test. Our algorithm exhibits superior test power (true-positive rate) when compared to prior methods, even at extremely low false-positive error rates (as low as 0). Also, under computation constraints, where only a limited number of reference models (as few as 1) are available, our method performs exceptionally well, unlike some prior attacks that approach random guessing in such scenarios. Our method lays the groundwork for cost-effective and practical yet powerful and robust privacy risk analysis of machine learning algorithms.

## 1 INTRODUCTION

Membership inference attacks (MIA) determine whether a specific data point has been used in training of a model (Shokri et al., 2017). These attacks represent a foundational tool in evaluating the privacy risks of unintentional exposure of information due to training machine learning models on different types of data in a wide range of scenarios. These scenarios encompass diverse settings such as statistical models (Homer et al., 2008; Backes et al., 2016; Sankararaman et al., 2009; Murakonda et al., 2021), machine learning as a service (Shokri et al., 2017), federated learning (Nasr et al., 2019; Li et al., 2023; Jagielski et al., 2023), generative models (Carlini et al., 2021), and also privacy-preserving machine-learning (Steinke et al., 2023; Nasr et al., 2021; Jagielski et al., 2020). Membership inference attacks originated within the realm of summary statistics on high-dimensional data (Homer et al., 2008). In this context, multiple hypothesis testing methods were developed to optimize the trade-off between test power and associated errors for relatively straightforward computations (Sankararaman et al., 2009; Dwork et al., 2015; Murakonda et al., 2021). For deep learning algorithms, these tests evolved from using machine learning itself to perform the membership inference test (Shokri et al., 2017) to using various approximations of the original statistical tests (Sablayrolles et al., 2019; Ye et al., 2022; Carlini et al., 2022; Watson et al., 2022b). They also vary based on the assumptions about threat models, as well as the amount of computation needed to tailor the attacks to specific data points and models (e.g., global attacks (Shokri et al., 2017; Yeom et al., 2018) versus per-sample tailored attacks (Ye et al., 2022; Carlini et al., 2022; Sablayrolles et al., 2019; Watson et al., 2022b)) which necessitate training a *large* number of reference models.

Even though there have been substantial improvements in the effectiveness of attacks, their **computational expense** has rendered them useless for practical privacy auditing. This is because the auditor would have to dedicate significantly more resources to performing the privacy test than they would to training the model itself. As we demonstrate in this paper, with a constrained computation budget, some of these attacks, e.g., (Carlini et al., 2022), verge on **random guessing**, for generalized models. Furthermore, prior state-of-the-art attacks (Ye et al., 2022; Carlini et al., 2022) use seemingly distinct likelihood ratio tests, and beyond empirical assessments, they do not provide a clear, *interpretable* means of comparison. Also, as evidenced both in their papers and reproduced in ours, these attacks exhibit mutual dominance, dominating each other depending on the test scenarios, such as variations in the number of reference models. This calls for **more robust yet efficient** attacks.

**We design robust attack algorithms that consistently achieve a high TPR on a limited computation budget (specifically given a few reference models), while maintaining effectiveness across all FPR, even as small as 0. Our attack (RMIA) dominates prior work in all scenarios.**

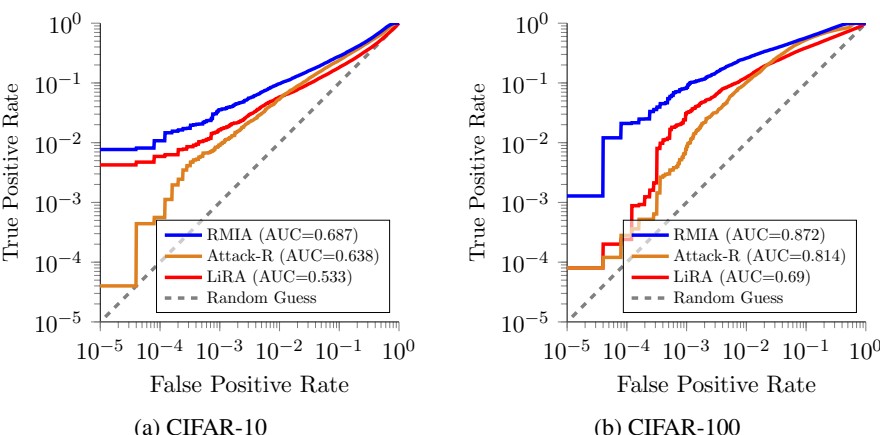

(a) CIFAR-10            (b) CIFAR-100

Figure 1: The performance comparison between our attack (RMIA) and the prior works (including Attack-R (Ye et al., 2022) and LiRA (Carlini et al., 2022)), under computation constraints, with the restriction of using only 1 reference model, for attacking one single model. LiRA approaches random guessing, and RMIA outperforms other attacks throughout the TPR-FPR trade-off curve.

Our method outperforms prior state-of-the-art attacks Attack-R (Ye et al., 2022) and LiRA (Carlini et al., 2022) across all datasets, by achieving $5-10\%$ higher AUC and a remarkable $2\times$ to $4\times$ higher TPR at low FPRs, when using 2 reference models. When considering just 1 single reference model, the improvement in AUC reaches $26\%$, compared with LiRA. See Figure 1. When dealing with a few reference models, Attack-R mainly suffers from low TPR at low FPR, while LiRA fails to get a competitive AUC score beyond random guessing. In an offline scenario where the adversary exclusively uses pre-trained reference models, RMIA demonstrates an impressive $28\%$ higher AUC and $3\times$ better TPR at **zero FPR** compared to LiRA. The offline version of our attack shows a comparable performance to online attacks, as we aim to avoid the huge cost associated with training online reference models. We also explore the effects of increasing available resources up to the levels used in the prior works. Even though the other methods show a reasonable performance when using large number (over 250) of reference models, our method still dominates them on benchmark datasets.

Our key **innovation** lies in designing a novel membership inference attack that effectively incorporates both population data as well as models trained on them as reference points in our hypothesis test. When modeling the hull hypothesis, the prior work mostly compute the *average* likelihood of the target data point not being included in the training set. To improve the power of the test, we distinguish between the worlds in which the target data point could have been replaced with any random sample from the population. Thus, the optimal adversary strategy calculates the likelihood ratio over the null hypothesis tied to random samples from the population. The computation of the likelihood ratio itself requires comparison with reference models. Essentially, **our membership inference test efficiently gauges the multiplicative distance between the probability of the target data point and a random sample from the population, when computed on the target model versus when it is computed from reference models**. Thus, our attack is finely calibrated to the interplay between data points and their *relative* probabilities in relation to models. This enhances the differentiation between member and non-member data points, enabling more precise estimation of test statistics and yielding a more resilient test. A significant aspect of our framework and attack is its foundational nature; prior attacks can be framed as simplifications of ours. Our interpretation of the prior work, using our framework, reveals the implicit assumptions and approximations in other methods, shedding light on their reduced performance and instability.

Through extensive empirical analysis on benchmark datasets, we investigate the impact of varying the number of models, the number of required inference queries, the similarity of reference models to the target model, and the parameters in the attack construction. In all these scenarios, we notice instabilities in the effectiveness of the prior attacks, depending on the settings. For example, in offline setting or when using a few reference models, Attack-R dominates LiRA, but when using many reference models in the online setting, LiRA outperforms Attack-R. However, even when considering worst-case scenarios, **RMIA consistently outperforms prior attacks in all settings**.

## 2 OUR METHOD

Membership inference attack (MIA) algorithms aim to determine whether a specific data point $x$ was used in the training of a given machine learning model $\theta$. The concept of a membership inference attack is modeled as an indistinguishability game between a challenger (the algorithm) and an adversary (the privacy auditor) (Ye et al., 2022; Carlini et al., 2022; Yeom et al., 2018). We present the standard MIA game in Definition 1. For a comprehensive understanding of membership inference games, see (Ye et al., 2022), and for their connection to other inference attack games, see (Salem et al., 2023). Essentially, there are two scenarios or worlds. In one world, the model $\theta$ is trained including $x$ in the training set, whereas in the other, it excludes $x$. The adversary is randomly positioned in one of these worlds and tasked with inferring which world he is in, using only data point $x$, the trained model $\theta$, and his background knowledge about the data distribution.

**Definition 1 (Membership Inference Game)**    *(Shokri et al., 2017; Yeom et al., 2018; Carlini et al., 2022; Ye et al., 2022) Let $\pi$ be the data distribution, and let $\mathcal{T}$ be the training algorithm.*

- *The challenger samples a training dataset $S \sim \pi$, and trains a model $\theta \sim \mathcal{T}(S)$.*

- *The challenger flips a fair coin $b$. If $b = 1$, it randomly samples a data point $x$ from $S$. Otherwise, it samples $x \sim \pi$, such that $x \notin S$. The challenger sends the target model $\theta$ and the target data point $x$ to the adversary.*[1]

- *The adversary, having access to the distribution over the population data $\pi$, outputs a membership prediction bit $\hat{b} \leftarrow \mathrm{MIA}(x; \theta)$.*

A membership inference attack is a hypothesis testing problem that assigns a membership score $\mathrm{Score}_{\mathrm{MIA}}(x; \theta)$ to every pair of $(x, \theta)$, and outputs a membership bit through comparing the score with a threshold (Yeom et al., 2018; Carlini et al., 2022; Ye et al., 2022):

$$\mathrm{MIA}(x; \theta) = \mathbb{1}_{\mathrm{Score}_{\mathrm{MIA}}(x;\theta) \geq \beta} \tag{1}$$

For any given threshold $\beta$, the adversary's **power**, defined as the true positive rate of the attack, and **error** or the false positive rate, are quantified over numerous repetitions of this experiment.[2] The threshold $\beta$ controls how much error the adversary is willing to tolerate. The universal goal for designing membership inference attacks is to maximize the adversary's power for any false-positive error rate. The (lower-bound) *leakage* of the algorithm is defined as the power-error trade-off curve (the ROC curve), which is derived from the outcome of game experiments across all values of $\beta$.

### 2.1 DESIGNING OUR MEMBERSHIP INFERENCE ATTACK

The key to the design of membership inference attacks is the formulation of the hypothesis test, the construction of the two types of worlds (where $x$ is member of the training set of $\theta$ in one, and is a non-member in the other), and the evaluation of the corresponding likelihood ratio tests.[3] Prior works simplify the construction of the worlds, which results in low-power, unstable, and average-case tests. Here, we design a fine-grained construction of the worlds in the following way.

We compose the null hypothesis as the worlds in which the target data point $x$ is replaced with a random data point $z$ from the population. Thus, we design many **pairwise likelihood ratio tests** to test the membership of a data point $x$ *relative* to other data point $z$. To reject the null hypothesis, we need to collect a significant amount of evidence that the probability of $\theta$ for $x$ being in the training set is larger than the probability of $\theta$ when instead a random $z$ is in the training set. This approach

---

[1]Note that both $S$ and $x$ can also be selected by the adversary to model the worst-case scenarios as described in the construction of MIA games (Ye et al., 2022) to reflect the maximum leakage corresponding to the differential privacy bound (Dwork et al., 2006). Our problem formulation and the tests can also apply to the cases where adversary controls sources of randomness in data sampling, but in the main text we focus on data points randomly sampled from the population.

[2]Power is the fraction of times $\hat{b} = 1$, given $b = 1$. Error is the fraction of times $\hat{b} = 1$, given $b = 0$.

[3]Likelihood ratio test is the best technique that the adversary can choose (Sankararaman et al., 2009; Murakonda et al., 2021; Ye et al., 2022; Carlini et al., 2022).

provides a much more fine-grained analysis of leakage, and differentiates between the worlds in which $x$ is not a member. The likelihood ratio corresponding to the pair of $x$ and $z$ is:

$$\text{LR}_\theta(x, z) = \frac{\Pr(\theta|x)}{\Pr(\theta|z)}, \tag{2}$$

where $\Pr(\theta|.)$ is computed over the randomness of the training algorithm (e.g., SGD). The term $\Pr(\theta|x)$ is the probability that the algorithm produces the model $\theta$ given that $x$ was in the training set, while the rest of the training set is randomly sampled from the population distribution $\pi$.

In the next subsection, we explain the process for computing $\text{LR}_\theta(x, z)$, which requires having access to reference models. Given $\text{LR}_\theta(x, z)$, we formulate the hypothesis test for our novel membership inference attack, which essentially is a test for violation of privacy, as follows:

$$\text{Score}_{\text{MIA}}(x; \theta) = \Pr_{z \sim \pi} \left( \text{LR}_\theta(x, z) \geq \gamma \right) \tag{3}$$

We measure the probability that $x$ can $\gamma$-*dominate* a random sample $z$ from the population. The threshold $\gamma \geq 1$ determines how much larger the probability of learning $\theta$ with $x$ as a training data should be *relative* to a random alternative point $z$ to pass our test. As $\gamma$ increases, the test looks for an evidence of high leakage, but the chance of finding such dominated $z$ samples decreases (especially when the model has generalized). The standard threshold $\beta$ in equation 1 specifies that there should be a sufficient fraction of the randomly sampled population data to affirm that $x$ is a member. By performing the test over $\beta \in [0, 1]$, we can compute the ROC power-error trade-off curve corresponding to the membership inference attack.

## 2.2 COMPUTING THE LIKELIHOOD RATIO

We can apply the Bayes rule to compute the likelihood ratio equation 2:

$$\text{LR}_\theta(x, z) = \left( \frac{\Pr(x|\theta)\,\cancel{\Pr(\theta)}}{\Pr(x)} \right) \cdot \left( \frac{\Pr(z|\theta)\,\cancel{\Pr(\theta)}}{\Pr(z)} \right)^{-1} = \left( \frac{\Pr(x|\theta)}{\Pr(x)} \right) \cdot \left( \frac{\Pr(z|\theta)}{\Pr(z)} \right)^{-1} \tag{4}$$

Here, $\Pr(x|\theta)$ is the likelihood function of model $\theta$ evaluated on data point $x$. In the case of classification models, usually the loss function is the negative log likelihood. So, $\Pr(x|\theta)$ is equivalent to the normalized score (SoftMax) of output of the model $f_\theta(x_{\text{features}})$ on class $x_{\text{label}}$ (MacKay, 2003; Blundell et al., 2015). See Appendix A.10 for better alternatives to SoftMax for computing $\Pr(x|\theta)$.

It is important to note that $\Pr(x)$ is not the same as $\pi(x)$, which is rather the prior distribution over $x$. The term $\Pr(x)$ is defined as the normalizing constant in the Bayes rule, and has to be computed by integrating over all models $\theta'$ with the same structure and training data distribution as $\theta$.

$$\Pr(x) = \sum_{\theta'} \Pr(x|\theta')\Pr(\theta') = \sum_{D,\theta'} \Pr(x|\theta')\Pr(\theta'|D)\Pr(D) \tag{5}$$

We compute $\Pr(x)$ as the empirical mean of $\Pr(x|\theta')$ by sampling *reference models* $\theta'$, each trained on random datasets $D$ drawn from the population distribution $\pi$. Note that the reference models must be sampled in an *unbiased* way with respect to whether $x$ is part of their training data. This is because the summation in equation 5 is over *all* $\theta'$, which can be partitioned to the set of models trained on $x$ (IN models) and the set of models that are *not* trained on $x$ (OUT models). See Appendix A.10.3, for the details of computing $\Pr(x)$ from IN and OUT models in the online attack setting, as well as its approximation in the offline setting where we only have OUT models. The same reasoning and computation process applies to $\Pr(z)$.

### 2.2.1 MEMBERSHIP INFERENCE ATTACK

Given the $\text{LR}_\theta(x, z)$ computation in equation 4, we compute the $\text{Score}_{\text{MIA}}(x; \theta)$ as in equation 3, and finally perform the membership inference test as in equation 1. Definition 2 presents our attack procedure corresponding to the MIA game in Definition 1 (we provide a detailed pseudo-code in Appendix A.1). We assume adversary has access to random samples from the population, and also some reference models.

**Definition 2 (Relative Membership Inference Attack – RMIA)**

- *Input: model $\theta$, data point $x$, and test parameters $\gamma$, $\beta$.*

- *Sample many $z \sim \pi$, and compute $\mathrm{Score}_{\mathrm{MIA}}(x; \theta)$ as the fraction of $z$ samples that pass the relative membership inference likelihood ratio test $\mathrm{LR}_\theta(x, z) \geq \gamma$. [See equation 3]*

- *Return* MEMBER *if $\mathrm{Score}_{\mathrm{MIA}}(x; \theta) \geq \beta$, and* NON-MEMBER *otherwise. [See equation 1]*

We can further enhance the effectiveness of our attack by augmenting the MIA query with multiple data samples that are similar to $x$ (Carlini et al., 2022; Choquette-Choo et al., 2021). These data samples can be simple transformations of $x$ (for example, using shift or rotation in case of image data). To consolidate the results in our multi-query setting, we use majority voting on the hypothesis test: $x$ is considered to dominate $z$ if more than half of all generated transformations of $x$ dominate $z$.

## 2.3 BOOSTED RELATIVITY OF RMIA, AND DESIGN IMPROVEMENTS OVER PRIOR ATTACKS

Membership inference attacks, framed as hypothesis tests, essentially compute the *relative* likelihood of observing $\theta$ given $x$'s membership in the training set of $\theta$ versus observing $\theta$ under $x$'s non-membership (null hypothesis). The key to a robust test is accounting for *all information sources* that distinguish these possible worlds. Membership inference attacks use *references* as anchors from the null hypothesis worlds, comparing the pair $(x, \theta)$ against them. Effectively designing the test involves leveraging all possible informative references, which could be either population data or models trained on it. Homer et al. (2008) and its follow up methods use population data as a reference, while Sankararaman et al. (2009) and its follow up methods use reference models trained on such data. Recent MIA methods have predominantly focused on using reference models. The *way* that such reference models are used matters a lot. As we show in our empirical evaluation, prior state-of-the-art attacks (Carlini et al., 2022; Ye et al., 2022) exhibit different behavior depending on the reference models (i.e., in different scenarios, they *dominate each other in opposing ways*). Also, even though they outperform attacks that are based on population data (e.g., the Attack-P formulation of (Homer et al., 2008)) by a large margin, they do not strictly dominate them on all membership inference queries (Ye et al., 2022). They, thus, fall short due to overlooking some type of relativity.

Table 1 summarizes the MIA scores of various attacks. Our method offers a fresh perspective on the problem. This approach leverages both population data and reference models, enhancing attack power and robustness against changes in adversary's background knowledge. Our likelihood ratio test, as defined in equation 3 and equation 4, effectively measures the distinguishability between $x$ and any $z$ based on the shifts in their probabilities when conditioned on $\theta$, through contrasting $\Pr(x|\theta)/\Pr(x)$ and $\Pr(z|\theta)/\Pr(z)$. Notably, a class of strong prior attacks utilizing reference models, especially as seen in (Ye et al., 2022) and similar attacks, essentially mimic this test but neglect the $\Pr(z|\theta)/\Pr(z)$ component. Their dependence on the *uncalibrated* magnitude of $\Pr(x|\theta)/\Pr(x)$ results in our attack surpassing them throughout the power-error (TPR-FPR) curve. Calibration by $z$ tells us if the magnitude of $\Pr(x|\theta)/\Pr(x)$ is significant (compared to non-members).

Another category of attacks (Carlini et al., 2022) also falter, missing the essential calibration of their test with population data. But, the weakness of these attacks is not limited to this. To provide a better comparison, let us first introduce an alternative method to compute our likelihood ratio equation 2.

In the black-box setting, the divergence between the two (numerator and denominator) distributions in a MIA likelihood ratio is a stationary point *when* the model is queried on the differing points (Ye et al., 2023). This has been the practice in MIA attacks. So, the best strategy to maximize the

| **Method** | RMIA | LiRA | Attack-R | Attack-P | Global |
|---|---|---|---|---|---|
| **MIA Score** | $\Pr_z\left(\frac{\Pr(\theta\|x)}{\Pr(\theta\|z)} \geq \gamma\right)$ | $\frac{\Pr(\theta\|x)}{\Pr(\theta\|\bar{x})}$ | $\Pr_{\theta'}\left(\frac{\Pr(x\|\theta)}{\Pr(x\|\theta')} \geq 1\right)$ | $\Pr_z\left(\frac{\Pr(x\|\theta)}{\Pr(z\|\theta)} \geq 1\right)$ | $\Pr(x\|\theta)$ |

Table 1: Computation of $\mathrm{Score}_{\mathrm{MIA}}(x; \theta)$ in different membership inference attacks (RMIA, this paper, versus LiRA (Carlini et al., 2022), Attack-R and Attack-P (Ye et al., 2022), and Global (Yeom et al., 2018)), where the notation $\bar{x}$ (for LiRA) represents the case where $x$ is not in the training set. The attack in all methods is $\mathrm{MIA}(x; \theta) = \mathbb{1}_{\mathrm{Score}_{\mathrm{MIA}}(x;\theta) \geq \beta}$ based on the game in Definition 1.

likelihood ratio in the black-box setting is to evaluate $f_\theta(x_{\text{features}})$ and $f_\theta(z_{\text{features}})$, where $f_\theta(.)$ is a classification model with parameters $\theta$. Thus, a **direct** way to compute equation 2 is the following:

$$\text{LR}_\theta(x, z) = \frac{\Pr(\theta|x)}{\Pr(\theta|z)} \approx \frac{\Pr(f_\theta(x), f_\theta(z)|x)}{\Pr(f_\theta(x), f_\theta(z)|z)}, \quad \text{``direct computation of LR''} \quad (6)$$

where the numerator $\Pr(f_\theta(x), f_\theta(z)|x)$ can be computed empirically by training many reference models $\theta'_x$ that are trained on $x$ and the rest of the training data are randomly sampled from the population. Similarly, for the denominator we need to train many reference models $\theta'_z$. Following Ye et al. (2023); Carlini et al. (2022), we can compute these probabilities using Gaussian distribution, on output (logits) of model at class $x_{\text{label}}$ when evaluated on $x_{\text{features}}$, as detailed in Appendix A.12. We provide an empirical comparison between attack performance of our main computation (equation 4 using Bayes rule) and direct computations of the likelihood ratio in Appendix A.12. The results show that both computations of our method match when we use a large number of reference models (Figure 20). However, our main computation using the Bayes rule equation 4 dominates the direct computation of LR equation 6 when a few reference models are used (Figure 21).

Given our equation 6, the LiRA test in (Carlini et al., 2022) can be viewed as an *average case* of our test. Observe that the denominator in LiRA LR is the average case for our pairwise LR averaged over all $z$. This reduces the power of LiRA's test. Also, as we show in Appendix A.12, a direct computation of LR requires a large number of reference models. Their LR numerator necessitates online training of IN reference models specific to each target query $x$, otherwise the attack performance is very low. Thus, our attack strictly dominates (Carlini et al., 2022) throughout the power-error (TPR-FPR) curve, and the gap increases significantly when we reduce the computation budget for reference models (See Figure 6). The combination of a pairwise LR and its computation using the Bayesian approach results in our robust, high-power, and low-cost attack.

Another way to interpret our test is by examining the relative distinguishability between $x$ and $z$ by comparing their probability ratios when evaluated using reference models versus their probability ratios under the target model. In other words, we contrast $\Pr(x|\theta)/\Pr(z|\theta)$ with $\Pr(z)/\Pr(x)$. If both points, $x$ and $z$, exhibit identical probability ratios when assessed against the target and reference models, they become indistinguishable, and any deviation from this is detected by the test. The strength of our test lies in its ability to detect subtle differences in these probability ratios stemming from inclusion of $x$ in the target training set. By repeatedly applying this RMIA test for numerous $z$ samples from the population, our membership inference attack gains a strong confidence in distinguishing members from non-members. In contrast, the prior work based on Homer et al. (2008) that solely depend on probability (or error) of the population data as characterized by Attack-P in (Ye et al., 2022), lack power (low TPR) due to their neglect of the $\Pr(z)/\Pr(x)$ component. Essentially, they operate under the inaccurate assumption that all data points are all alike, i.e., $\Pr(z) \approx \Pr(x)$.

## 3 EMPIRICAL EVALUATION

### 3.1 EXPERIMENTAL SETUP

Our evaluation is aimed at comparing the proposed attack with prior state-of-the-art membership inference attacks. For a better comparison, we use the same setup as (Carlini et al., 2022), in which for a given dataset, the adversary trains $k$ reference (shadow) models on training sets such that each sample $x \in \pi$ is contained in exactly half of the reference models' training set.

Here, we report the attack results on models trained with four different datasets, traditionally used for membership inference attack evaluations. For CIFAR-10 (He et al., 2016) (a traditional image classification dataset), we train a Wide ResNets (with depth 28 and width 2) to 92% test accuracy (for 100 epochs) on half of the dataset (25000 samples, chosen at random). For CIFAR-100 and CINIC-10 (as other image datasets), we follow the same process as for CIFAR-10 and train a wide ResNet on half of the dataset to get 67% and 77% test accuracy, respectively, surpassing the accuracy of models utilized in prior studies. We set the batch size to 256. We also include the result of attacks on Purchase-100 dataset (a tabular dataset of shopping records) (Shokri et al., 2017), where models are 4-layer MLP with layer units=[512, 256, 128, 64], trained on 25k samples for 50 epochs to obtain 83% test accuracy. We train our models using standard techniques to reduce over-fitting, including train-time augmentations, weight decay and early stopping. Exactly like (Carlini et al., 2022), there are a number of simple augmentations for each training sample in image models, computed by

horizontally flipping and/or shifting the image by a few pixels. As a result, the train-test accuracy gap of our models is small (e.g. below 7% for CIFAR-10 models).

We measure the performance of each attack using two underlying metrics: its true positive rate (TPR), and its false positive rate (FPR), over all member and non-member records of random target models. Then, we use the ROC curve to reflect the trade-off between the TPR and FPR of an attack, as we sweep over all possible values of threshold $\beta$ to build different FPR tolerance. The AUC (area under the ROC curve) score gives us the average success across all target samples and measures the overall strength of an attack. Inspired from previous discussions in (Carlini et al., 2022), we also consider TPR at very low FPRs. More precisely, we focus on TPR at 0% FPR, a metric that has seen limited usage in the literature. All samples in the population data are used as input queries. Hence, for each target model, half of queries are members and the other half are non-members.

### 3.2 COMPARISON OF DIFFERENT ATTACKS

We compare the performance of RMIA with three recent effective attacks, namely Attack-R and Attack-P introduced in (Ye et al., 2022) and LiRA in (Carlini et al., 2022). They have been shown to exhibit the best performance, compared to previous attacks, hence we do not bring the result of other attacks here. We can improve attacks by making multiple queries to the model. Therefore, in addition to querying the target model with query $x$, we also query on augmentations of $x$, obtained via simple mirror and shift operations. We apply the idea of augmented queries only on LiRA and RMIA, as the other two attacks do not originally support multiple queries.

We evaluate attacks under two general settings: 1) The offline setting where the adversary only trains reference models (or OUT models) that do not contain any input query in their training set, and 2) The online setting where the adversary is given enough time and resource to train reference models (or IN models) using input queries. The Attack-R is inherently an offline attack and the Attack-P is independent of reference models. In (Carlini et al., 2022), the authors described how to convert LiRA to an offline attack that only works with OUT models. Likewise, we can restrict our attack to utilize OUT models for computing $\Pr(x)$ in the likelihood ratio of equation 4, rendering it as an offline attack. In Appendix A.10.3, we also elaborate on improving the approximation of $\Pr(x)$ when dealing with a limited number of OUT models.

The main questions we are going to answer in this section are the followings:

1. How do attacks perform well when the adversary is able to train and use only a couple of reference models? This is an essential issue when working with models that need lots of data and huge amount of computation and memory for their training.

2. Which attack provides the best performance in an offline scenario where the adversary cannot train any model on receiving an input query? This scenario plays an important role in making an MIA algorithm useful for a practical privacy risk analysis task, as the cost and the time of training new models for each query is highly restrictive in many real cases.

3. Which attack gives the best result when there is no limitation for training reference models?

**Performance of Attacks under Limited Number of Reference Models.** Table 2 compares the result of attacks when we train a limited number of reference models, with CIFAR-10, CIFAR-100 and CINIC-10 datasets. We fix the total number of models used throughout the whole experiment such that the same set of models are used to infer the membership of all target samples. We believe such a constrained setting is more practical for privacy auditing tasks. When evaluating the offline version of LiRA and RMIA and also Attack-R, all reference models are OUT for each target sample. When the number of reference models is 1, we are only able to assess offline attacks. For online attacks (i.e. LiRA and RMIA), we train models in a way that half of them are IN and half are OUT for each target sample. We ignore to bring the result of the Attack-P for now, as it does not make use of reference models. The proposed RMIA can demonstrate its strict dominance using a few reference models across all datasets. For instance, with 2 CIFAR-10 models, it achieves around 10% higher AUC than both Attack-R and LiRA and still gains at least 110% better TPR at zero FPR. Surprisingly, the offline RMIA demonstrates an enhanced level of superiority over other attacks, including the online LiRA algorithm. For example, with 4 CIFAR-10 models, it has at least 6% higher AUC and a stunning 3x improvement for TPR at zero FPR over online LiRA. In the extreme case of using one reference model, RMIA shows at least 26% higher AUC and 100% more

| # Ref. Models | Attack | CIFAR-10 | | | CIFAR-100 | | | CINIC-10 | | |
|---|---|---|---|---|---|---|---|---|---|---|
| | | AUC | TPR@FPR | | AUC | TPR@FPR | | AUC | TPR@FPR | |
| | | | 0.01% | 0.0% | | 0.01% | 0.0% | | 0.01% | 0.0% |
| 1 | Attack-R | 63.65 ± 0.27 | 0.07 | 0.02 | 81.61 ± 0.17 | 0.06 | 0.02 | 72.04 ± 0.35 | 0.07 | 0.02 |
| | LiRA (Offline) | 53.20 ± 0.23 | 0.48 | 0.25 | 68.95 ± 0.28 | 0.54 | 0.27 | 59.93 ± 0.40 | 0.32 | 0.07 |
| | **RMIA (Offline)** | **68.64 ± 0.43** | **1.19** | **0.51** | **87.18 ± 0.14** | **2.06** | **0.77** | **79.00 ± 0.29** | **0.86** | **0.31** |
| 2 | Attack-R | 63.35 ± 0.30 | 0.32 | 0.08 | 81.52 ± 0.21 | 0.31 | 0.06 | 72.02 ± 0.32 | 0.21 | 0.07 |
| | LiRA (Offline) | 54.42 ± 0.34 | 0.67 | 0.27 | 72.21 ± 0.28 | 1.52 | 0.76 | 62.18 ± 0.47 | 0.57 | 0.26 |
| | **RMIA (Offline)** | **70.13 ± 0.37** | **1.71** | **0.91** | **88.92 ± 0.20** | **4.90** | **1.73** | **80.56 ± 0.29** | **2.14** | **0.98** |
| | LiRA | 63.97 ± 0.35 | 0.76 | 0.43 | 84.55 ± 0.16 | 1.15 | 0.55 | 73.17 ± 0.29 | 0.53 | 0.12 |
| | **RMIA** | 70.10 ± 0.46 | 0.98 | 0.54 | 88.76 ± 0.14 | 3.33 | 1.59 | 79.90 ± 0.34 | 1.05 | 0.50 |
| 4 | Attack-R | 63.52 ± 0.29 | 0.65 | 0.21 | 81.78 ± 0.19 | 0.63 | 0.19 | 72.18 ± 0.27 | 0.40 | 0.14 |
| | LiRA (Offline) | 54.60 ± 0.25 | 0.97 | 0.57 | 73.57 ± 0.31 | 2.26 | 1.14 | 63.07 ± 0.41 | 1.03 | 0.45 |
| | **RMIA (Offline)** | 71.02 ± 0.37 | **2.91** | **2.13** | 89.81 ± 0.17 | **7.05** | **3.50** | **81.46 ± 0.31** | **3.20** | **1.39** |
| | LiRA | 67.00 ± 0.33 | 1.38 | 0.51 | 87.82 ± 0.20 | 3.64 | 2.19 | 77.06 ± 0.29 | 1.34 | 0.51 |
| | **RMIA** | **71.25 ± 0.41** | 1.96 | 0.96 | **89.89 ± 0.19** | 4.95 | 2.44 | 81.16 ± 0.33 | 2.63 | 1.36 |

Table 2: Performance of attacks when a limited number of reference models are used. Separate models are trained with CIFAR-10, CIFAR-100 and CINIC-10 datasets. For LiRA (Carlini et al., 2022) and RMIA, we use 18 augmented queries, and for Attack-R (Ye et al., 2022), we use 1 query. For RMIA, we use $\gamma = 2$. Results are averaged over 10 random target models.

TPR at low FPRs than LiRA over all datasets. On the other hand, although the Attack-R shows a relatively high AUC, but it can never get a good TPR at lower FPRs. The reason is that it tries to predict the membership of a target sample just through comparing its loss in various reference models which is restrictive when we do not have enough models. Since RMIA takes the advantage of two information sources (reference records and models), it can better tolerate with fewer models.

We illustrate the ROC of these attacks on a random target model in Figure 1 in which the number of reference models is limited to 1. Figure 1a) and 1b) are obtained with models trained on CIFAR-10 and CIFAR-100, respectively. In both ROCs, the Attack-R and LiRA lag behind RMIA across nearly all FPR values. In Appendix A.2, we show the ROC of attacks using various number of reference models trained on different datasets. Furthermore, for a deeper understanding of attacks' behaviour, we examine the variation in MIA scores among different attacks in Appendix A.11.

**Evaluation of Attacks with Offline Reference Models.** Now, we compare the performance of offline attacks and also examine how they work with using more augmented queries. Note that Attack-P and Attack-R has no result for multiple queries. In this experiment, we use 127 OUT models for each sample. As shown in the Offline column of Table 3, RMIA outperforms LiRA by 28% higher AUC and a remarkable 3 times better TPR at zero FPR (when comparing the best result of two attacks). As we increase queries, RMIA gets a better result (for example, a 4x improvement in TPR at zero FPR and also about 4.6% higher AUC as queries go from 1 to 50), while LiRA cannot benefit from the advantage of more queries to improve its AUC. Note that we use the same technique to generate augmentations, as proposed in (Carlini et al., 2022). The Attack-P fails to achieve a good TPR at low FPRs. Specifically, it may misclassify a typical non-member sample, one with a high prediction probability in reference models, as a member and conversely, mistakenly classify an atypical member sample, one with a low prediction probability in reference models, as a non-member. Additionally, the Attack-R may wrongly label a high-quality member sample, one with a higher prediction probability compared to other samples in the population, as a non-member solely because of its higher probability in reference models. In contrast, RMIA is designed to overcome these limitations by considering both the characteristics of the target sample within reference models and its relative probability among other reference records. With no additional queries, RMIA presents a clear advantage over Attack-R (with 6.7% higher AUC and 116% more TPR at zero FPR) and the performance gap between two attacks widens with using more queries. Taking the high cost of training more models into account even in the case of offline scenario, it is fascinating that the result of offline RMIA with a few models, shown in Table 2, is close in terms of AUC to its result with 127 models. As shown in Appendix A.3, a consistent pattern of results is observed in offline attacks in the presence of data distribution shift, i.e. when the target models are trained on datasets different from those used for the reference models.

**Evaluation of Attacks with Abundant Reference Models**. In the Online column of Table 3, we show the result of LiRA and RMIA when all 254 IN and OUT reference models are available to the adversary. We also present the impact of using different number of augmented queries on the

| # Queries | Attack | Online | | | Offline | | |
|---|---|---|---|---|---|---|---|
| | | AUC | TPR@FPR | | AUC | TPR@FPR | |
| | | | 0.01% | 0.0% | | 0.01% | 0.0% |
| 1 | Attack-R | - | - | - | 64.41 ± 0.41 | 1.52 | 0.80 |
| | Attack-P | - | - | - | 58.19 ± 0.33 | 0.01 | 0.00 |
| | LiRA | 68.92 ± 0.42 | 1.78 | 0.92 | 56.12 ± 0.41 | 0.46 | 0.28 |
| | **RMIA** | **69.15 ± 0.35** | **2.26** | **1.80** | **68.74 ± 0.34** | **2.42** | **1.73** |
| 2 | LiRA | 71.28 ± 0.46 | 2.83 | 1.73 | 55.77 ± 0.46 | 1.16 | 0.59 |
| | **RMIA** | **71.46 ± 0.43** | **3.69** | **2.55** | **71.06 ± 0.39** | **3.64** | **2.46** |
| 18 | LiRA | 72.04 ± 0.47 | 3.39 | 2.01 | 55.18 ± 0.37 | 1.37 | 0.72 |
| | **RMIA** | **72.25 ± 0.46** | **4.31** | **3.15** | **71.71 ± 0.43** | **4.18** | **3.14** |
| 50 | LiRA | 72.26 ± 0.47 | 3.54 | 2.19 | 55.00 ± 0.36 | 1.52 | 0.75 |
| | **RMIA** | **72.51 ± 0.46** | **4.47** | **3.25** | **71.95 ± 0.44** | **4.39** | **3.22** |

Table 3: Performance of attacks when we use different number of augmented queries for LiRA (Carlini et al., 2022) and RMIA. We evaluate attacks in two different settings, shown in Online and Offline columns. In the online setting, we use 254 models where half of them are IN models and half are OUT (for each sample). The offline setting uses only 127 OUT models. Both Attack-P and Attack-R (Ye et al., 2022) do not originally support multiple augmented queries. The Attack-P does not work with reference models, thus we consider it offline. Models are trained with CIFAR-10. For RMIA, we use $\gamma = 2$. Results are averaged over 10 random target models.

| Attack | CIFAR-10 | | | CIFAR-100 | | | CINIC-10 | | | Purchase-100 | | |
|---|---|---|---|---|---|---|---|---|---|---|---|---|
| | AUC | TPR@FPR | | AUC | TPR@FPR | | AUC | TPR@FPR | | AUC | TPR@FPR | |
| | | 0.01% | 0.0% | | 0.01% | 0.0% | | 0.01% | 0.0% | | 0.01% | 0.0% |
| Attack-P | 58.19 ± 0.33 | 0.01 | 0.00 | 75.91 ± 0.36 | 0.01 | 0.00 | 66.91 ± 0.30 | 0.01 | 0.00 | 66.63 ± 0.27 | 0.01 | 0.00 |
| Attack-R | 64.41 ± 0.41 | 1.52 | 0.80 | 83.37 ± 0.24 | 4.80 | 2.59 | 73.64 ± 0.34 | 2.17 | 1.12 | 77.80 ± 0.41 | 1.02 | 0.42 |
| LiRA | 72.04 ± 0.47 | 3.39 | 2.01 | **91.48 ± 0.16** | 10.85 | 7.13 | 82.44 ± 0.30 | 4.43 | 2.92 | 83.23 ± 0.37 | 1.99 | 0.82 |
| **RMIA** | **72.25 ± 0.46** | **4.31** | **3.15** | 91.01 ± 0.14 | **11.35** | **7.78** | **82.70 ± 0.35** | **6.77** | **4.43** | **83.90 ± 0.36** | **2.56** | **1.29** |
| LiRA (Offline) | 55.18 ± 0.37 | 1.37 | 0.72 | 75.78 ± 0.33 | 2.53 | 1.13 | 64.51 ± 0.51 | 1.33 | 0.60 | 65.82 ± 0.58 | 0.31 | 0.11 |
| **RMIA (Offline)** | **71.71 ± 0.43** | **4.18** | **3.14** | **90.57 ± 0.15** | **11.45** | **6.16** | **82.33 ± 0.32** | **5.07** | **3.33** | **83.21 ± 0.33** | **2.35** | **0.69** |

Table 4: Performance of different attacks on CIFAR-10, CIFAR-100, CINIC-10 and Purchase-100 datasets using 254 reference models. For LiRA (Carlini et al., 2022) and RMIA, We use 18 augmented queries, and for Attack-P and Attack-R (Ye et al., 2022), we use 1 query. For RMIA, we use $\gamma = 2$. Results are averaged over 10 random target models.

performance of two attacks. Compared with LiRA, RMIA always has a slightly higher AUC and at least 48% better TPR at zero FPR. Note that in this case, even minor AUC improvements are particularly significant when closely approaching the true leakage of the training algorithm through hundreds of models. Both LiRA and RMIA work better with increasing augmented queries, e.g. around 2x improvement in TPR@FPR when going from 1 query to 50 queries. Table 4 presents the performance of all attacks on models trained with four different datasets. In this experiment, we use all 254 reference models (for offline attacks, we use half of them). We observe roughly the same order of supremacy between attacks in all datasets. RMIA works slightly better than LiRA with respect to AUC, except for CIFAR-100. But, its TPR at zero FPR is improved considerably (by up to 50%) in various datasets. In addition, our offline attack consistently outperforms offline LiRA by at least 20% in AUC and 300% in TPR@FPR across all datasets. In fact, it demonstrates a performance comparable to online attacks which is quite remarkable, when we take into account the training costs associated with online models.

In the appendix, we study the impact of using different network architectures (Appendix A.4), DP-SGD (Appendix A.5) and also other ML algorithms (Appendix A.6) on the performance of attacks. In addition, we discuss how the number of reference records and also the selection of $\gamma$ and $\beta$ parameters affect RMIA in Appendix A.8 and A.9, respectively.

## 4 CONCLUSIONS

We design novel membership inference attacks, focusing on maximizing the distinguishability between members of a model's training set and random samples from the population, while minimizing computation costs. Our comprehensive evaluations across different settings shows the clear advantages of RMIA over the prior state-of-the-art attacks (Ye et al., 2022; Carlini et al., 2022), regardless of the dataset, number and quality of reference models, and training algorithms.

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

CONTENTS

LIST OF TABLES

## LIST OF FIGURES

# A  SUPPLEMENTARY EMPIRICAL RESULTS

## A.1  DETAILS OF RMIA AND ITS EVALUATION

In this section, we first explain how we evaluate different membership inference attack (MIA) algorithms and then describe the specifics of our proposed RMIA , elucidating its implementation through pseudo-code. Algorithm 1 outlines the general scheme frequently used to evaluate MIA algorithms. Given a target model $\theta$ and its training set $D_\theta$, the algorithm first generates the set of test queries (line 1), denoted by $Q$, by randomly and equally selecting members and non-members of $\theta$. Subsequently, the algorithm iterates over the target (test) samples in $Q$ (line 3) and computes a MIA score ($\text{Score}_{\text{MIA}}(x; \theta)$) for each target sample $x$ in line 4. This computation is done by invoking the COMPUTE_MIA_SCORE function of the under-studied MIA algorithm. For shadow model-based attacks such as LiRA (Carlini et al., 2022), Attack-R (Ye et al., 2022), and RMIA, this function trains and uses a set of reference models to compute the score for each input query. We can easily parallelize each attack across multiple target samples by training reference models for all test samples simultaneously, rather than training per test sample.

As indicated in equation 1, the membership status of a target sample $x$ is determined by comparing the assigned MIA score with a threshold, denoted as $\beta$ (line 5). If the score is equal to or greater than $\beta$, the sample is classified as a member; otherwise, it is classified as a non-member. The parameter $\beta$ is a common element in all attacks, and its value in each attack is determined empirically with the goal of maximizing the true positive rate (TPR) at a specified false positive rate (FPR) value. Correctly inferring the membership of a sample $x$ in line 6 results in an increase in the number of true positives. Conversely, incorrect inference of non-member samples (line 8) leads to an increase in false positives. Finally, we calculate the TPR and FPR by normalizing the number of true positives and false positives, respectively. In order to calculate the ROC curve and the corresponding area under the ROC curve (AUC), we sweep over all possible values of $\beta$ to build different FPR tolerances and run Algorithm 1 for each value. The final ROC curve is then derived from the obtained list of TPR and FPR values.

---

**Algorithm 1 General Scheme for Evaluating Membership Inference Attack Algorithms**.

---

**Require:**  target model $\theta$ with training set $D_\theta$
1: Select two subsets of equal size: $Q_{mem}$ which is randomly sampled from the training set $D_\theta$, and $Q_{non}$ randomly sampled from $\pi$ such that $Q_{non} \cap D_\theta = \varnothing$. The combined set, namely $Q = Q_{mem} \cup Q_{non}$, is then used as the test query set.          ▷ Generate test queries
2: $TP, FP \leftarrow 0, 0$
3: **for** each sample $x$ in $Q$ **do**
4:     $\text{Score}_{\text{MIA}}(x; \theta) \leftarrow \text{COMPUTE\_MIA\_SCORE}(\theta, x, ...)$      ▷ Compute MIA score of sample $x$
5:     **if** $\text{Score}_{\text{MIA}}(x; \theta) \geq \beta$ **then**          ▷ Predicted as member, based on equation 1
6:         **if** $x \in Q_{mem}$ **then**
7:             $TP \leftarrow TP + 1$                 ▷ Correctly predicted member (True Positive)
8:         **else if** $x \in Q_{non}$ **then**
9:             $FP \leftarrow FP + 1$           ▷ Incorrectly predicted non-member (False Positive)
10: $TPR \leftarrow \frac{TP}{|Q_{mem}|}$
11: $FPR \leftarrow \frac{FP}{|Q_{non}|}$

---

Algorithm 2 shows the pseudo-code of online RMIA. Initially, we select our reference samples (i.e., the set $Z$) from the population data $\pi$ and proceed to train $k$ reference models on random samples from the population. Each sample in $Z \cup x$ contributes to the training of $k/2$ models, categorized as IN models and OUT models, respectively. Subsequently, we query the target and reference models, calculating the ratio of prediction probabilities between the target model and the reference models for the target sample $x$ (line 5). When querying a sample on any model, we use Softmax or its alternatives as the prediction likelihood function of the model on the given sample. We repeat this ratio computation for each reference sample $z \in Z$ (line 7). As formulated in equation 4, the division of the computed ratio for the target sample $x$ by the obtained ratio for the reference sample $z$ determines $\text{LR}_\theta(x, z)$ to assess if $z$ is $\gamma$-dominated by $x$ (line 8). The fraction of $\gamma$-dominated reference samples in line 10 establishes the MIA score ($\text{Score}_{\text{MIA}}(x; \theta)$) of the target sample $x$ in RMIA. The final membership decision is made by comparing this score with the

threshold $\beta$ (line 5 in Algorithm 1). The offline version of RMIA pre-trains models on randomly sampled datasets in advance, avoiding any training on target samples. The modification required in Algorithm 2 to incorporate the offline RMIA is the exclusion of training any IN models in line 3. Another modification is that the denominator in ratios $Ratio_x$ and $Ratio_z$ (line 5 and 7, respectively) need to be computed according to the method in Appendix A.10.3, where we approximate what the denominators would have been if we had had access to IN models.

---

**Algorithm 2 MIA Score Computation with RMIA** . The function takes the target model $\theta$, the target (test) sample $x$, and the $\gamma$ parameter as inputs.

---

1: **function** COMPUTE_MIA_SCORE($\theta$, $x$, $\gamma$)
2:      Randomly choose a subset $Z$ from the population data $\pi$.      ▷ Choose reference samples
3:      Train $k$ models such that each sample $s \in Z \cup x$ is included for training $k/2$ models and excluded from the other $k/2$ models. $\Theta$ is the set of all models.      ▷ Train IN and OUT models
4:      $C \leftarrow 0$      ▷ Number of $\gamma$-dominated reference samples
5:      $Ratio_x \leftarrow \frac{\Pr(x|\theta)}{\sum_{\theta' \in \Theta} \Pr(x|\theta')}$      ▷ Query target and reference models on target sample $x$
6:      **for** each sample $z$ in $Z$ **do**
7:          $Ratio_z \leftarrow \frac{\Pr(z|\theta)}{\sum_{\theta' \in \Theta} \Pr(z|\theta')}$      ▷ Query target and reference models on reference sample $z$
8:          **if** $(Ratio_x/Ratio_z) > \gamma$ **then**      ▷ Compute $\mathrm{LR}_\theta(x, z)$, as in equation 4
9:              $C \leftarrow C + 1$
10:      $\mathrm{Score}_{\mathrm{MIA}}(x; \theta) \leftarrow C/|Z|$      ▷ Fraction of $\gamma$-dominated reference samples as MIA score
11:      **return** $\mathrm{Score}_{\mathrm{MIA}}(x; \theta)$

---

## A.2 ROCs Obtained on Different Datasets

We here compare the ROC of attacks using reference models trained on four different datasets, i.e. CIFAR-10, CIFAR-100, CINIC-10 and Purchase-100. We evaluate attacks when different number of reference models is used. To facilitate a more comprehensive comparison of the overall performance between various attacks, we provide a depiction of the ROC curves in both log and normal scales. In Figure 2, we show the ROC of attacks obtained with using 1 reference model. Since we only have 1 (OUT) model, the result of offline attacks is reported here. RMIA works remarkably better than other three attacks across all datasets. Although LiRA has a rather comparable TPR at low FPR for CIFAR-10 and Purchase-100 models, but it yields a much lower AUC (e.g. 22% lower AUC in CIFAR-10), when compared with RMIA. In other two datasets, RMIA results in around 3 times better TPR at zero FPR, than its closest rival.

Figure 3 displays the ROC of attacks resulted from employing 2 reference models. For each target sample, we use 1 IN and 1 OUT model. Again, RMIA works much better than other three attacks (in terms of both AUC and TPR@FPR) across all datasets. For example, it has a 10% higher AUC in CIFAR-10 models and at least 3 times better TPR at zero FPR in CIFAR-100 and CINIC-10, than other attacks. Figure 4 presents the ROC of offline attacks when using 127 OUT models. RMIA works much better than other three attacks across all datasets. For example, it leads to at least 20% higher AUC than LiRA . Figure 5 illustrates the ROC of attacks obtained when using all 254 models. For each target sample, we use 127 IN and 127 OUT models. With the help of training hundreds of IN and OUT models, LiRA can work close to our attack in terms of AUC, but the TPR at zero FPR of RMIA is considerably higher, e.g. by at least 50% in CIFAR-10, CINIC-10 and Purchase-100 models, as compared with LiRA .

For a better comparison between attack performances concerning the number of reference models, Figure 6 presents the AUC results for both offline and online attacks across varying reference model counts (ranging from 1 to 254). The left plots in this figure showcase the outcomes of offline attacks, while the right plots highlight the performance of online attacks. A consistent trend emerges, revealing that an increase in the number of reference models yields an improvement in AUC across all attacks. Notably, in both offline and online scenarios and across all datasets, RMIA consistently outperforms other attacks, particularly when employing a limited number of models. Intriguingly, the results from our offline RMIA, using a small number of reference models (e.g., 4), nearly align with the best outcomes achieved by online algorithms working with hundreds of models.

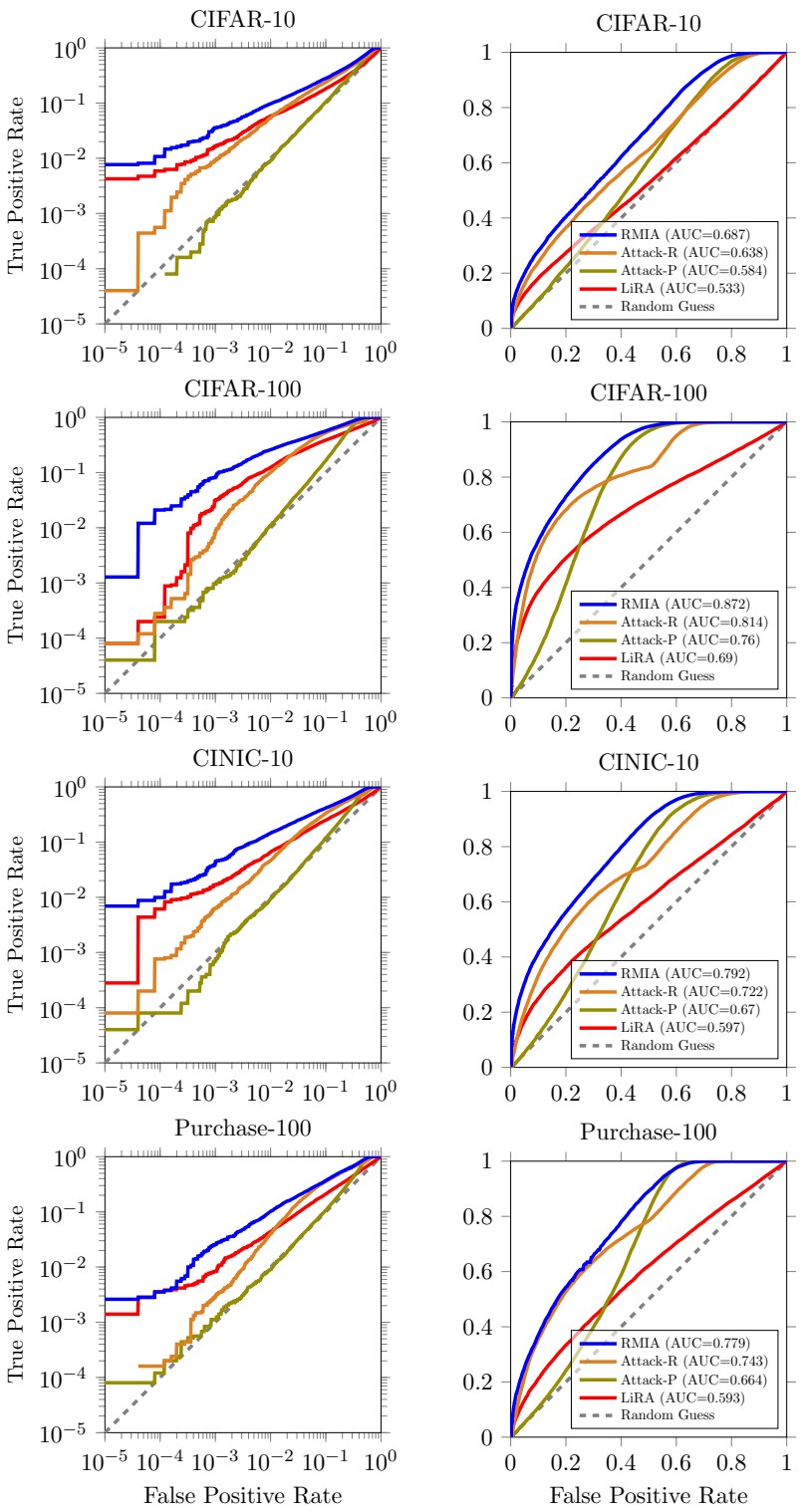

Figure 2: The ROC of attacks using models trained on different datasets (ROCs are shown in both log and normal scales). The result is obtained on one random target model. We here use **1 reference model** (OUT) for each target sample. In LiRA (Carlini et al., 2022) and RMIA, we use 18 augmented queries and for Attack-R and Attack-P (Ye et al., 2022), we use 1 query. For RMIA, we use $\gamma = 2$.

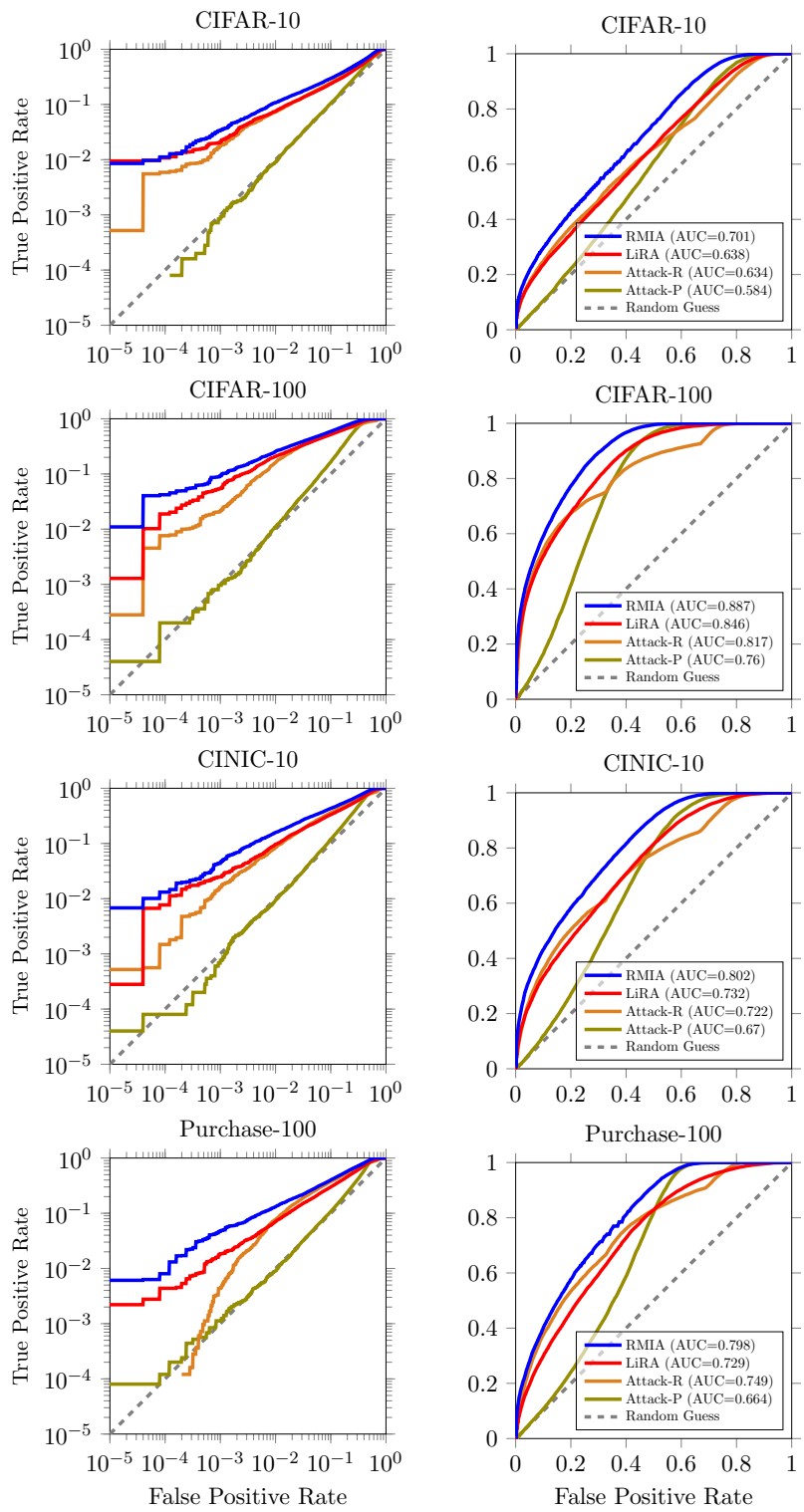

Figure 3: The ROC of attacks using models trained on different datasets (ROCs are shown in both log and normal scales). The result is obtained on one random target model. We here use **2 reference models** (1 IN and 1 OUT) for each target sample. In LiRA (Carlini et al., 2022) and RMIA, we use 18 augmented queries and for Attack-R and Attack-P (Ye et al., 2022), we use 1 query. For RMIA, we use $\gamma = 2$.

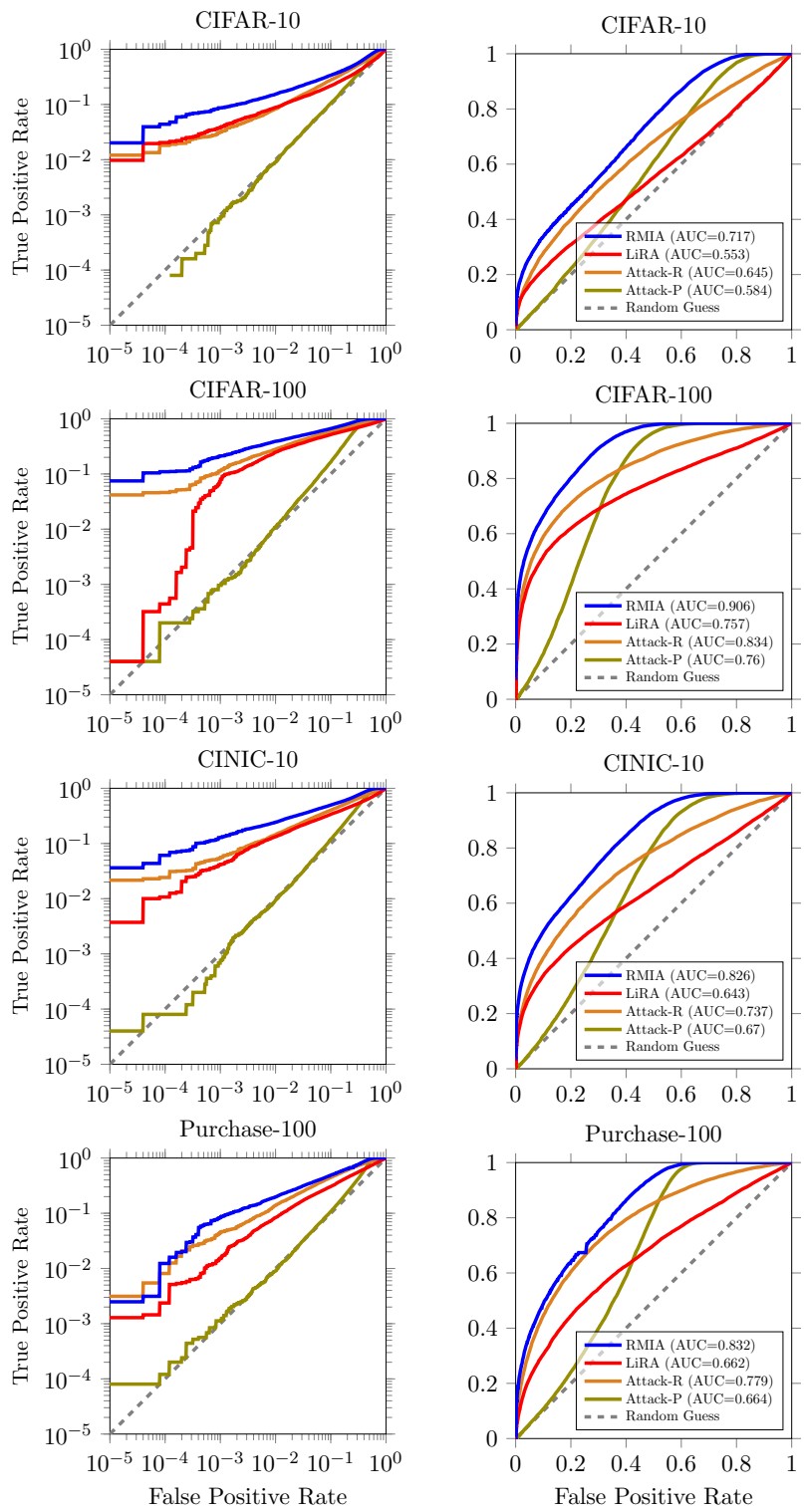

Figure 4: The ROC of offline attacks using models trained on different datasets (ROCs are shown in both log and normal scales). The result is obtained on one random target model. We use **127 OUT reference models** for each target sample. In LiRA (Carlini et al., 2022) and RMIA, we use 18 augmented queries and for Attack-R and Attack-P (Ye et al., 2022), we use 1 query. For RMIA, we use $\gamma = 2$.

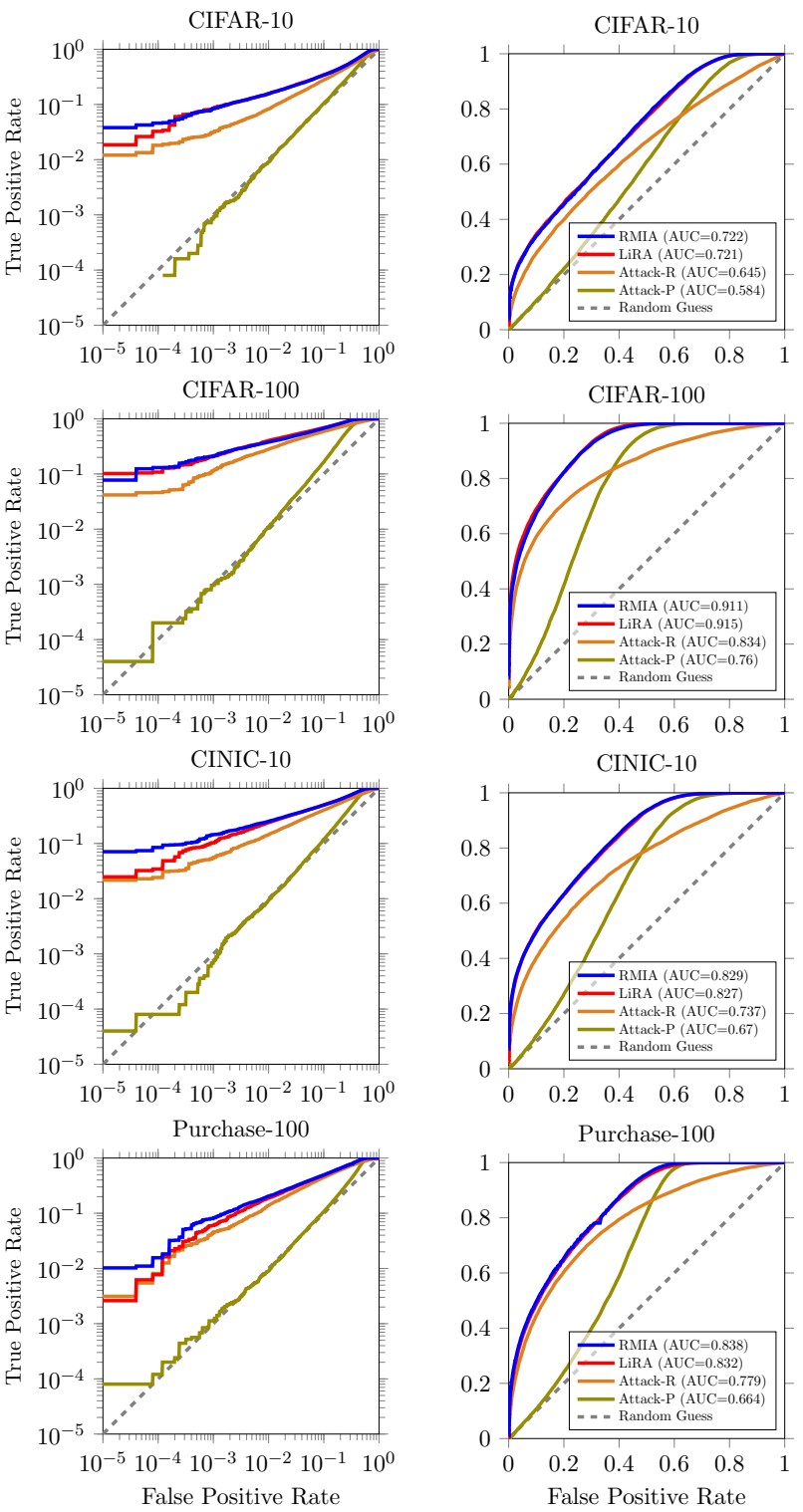

Figure 5: The ROC of attacks using models trained on different datasets (ROCs are shown in both log and normal scales). The result is obtained on one random target model. We here use **254 reference models** (127 IN and 127 OUT) for each target sample. In LiRA (Carlini et al., 2022) and RMIA, we use 18 augmented queries and for Attack-R and Attack-P (Ye et al., 2022), we use 1 query. For RMIA, we use $\gamma = 2$.

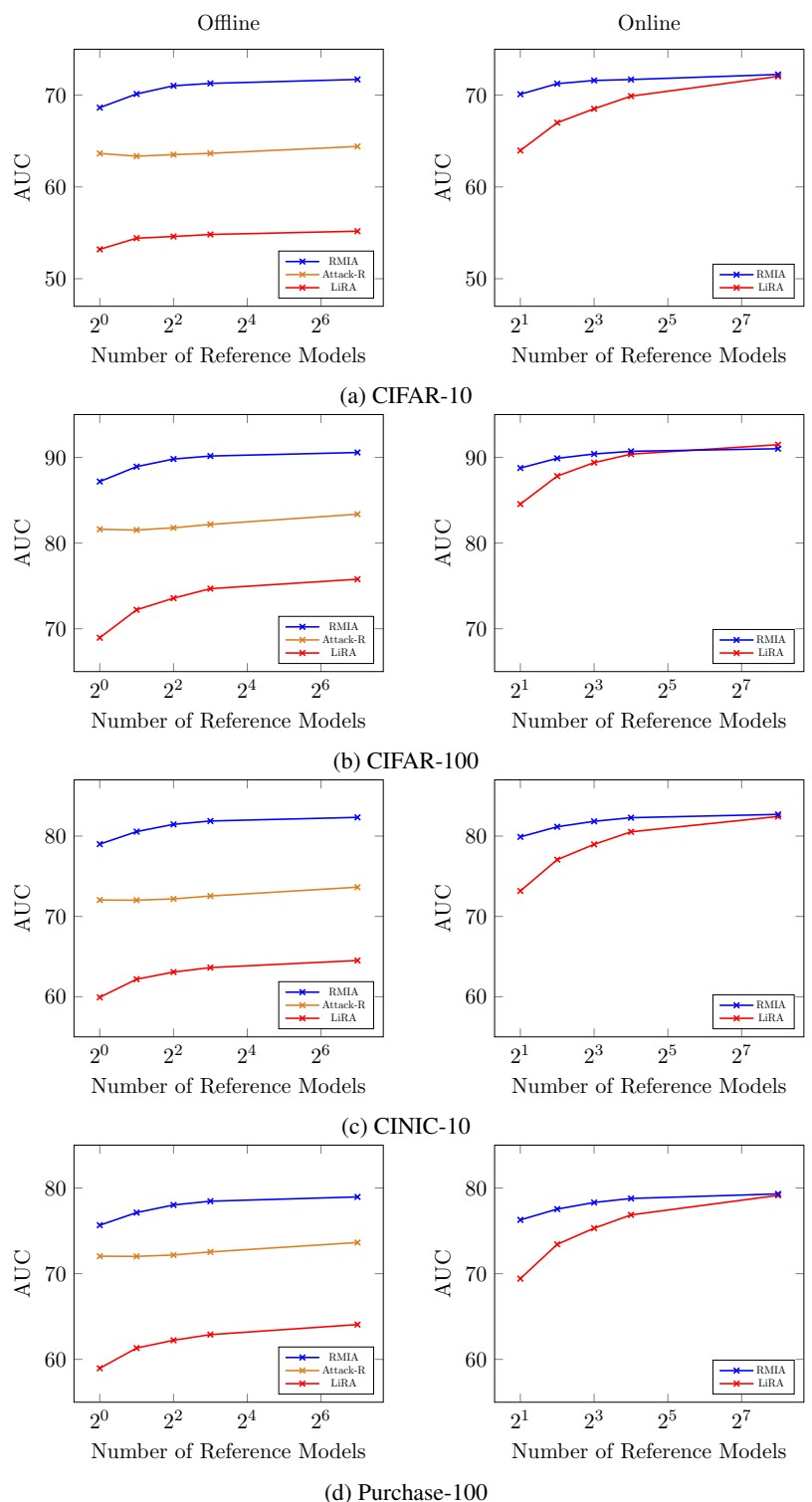

Figure 6: The AUC of various attacks obtained with using different number of reference models. The left plots illustrate the results of offline attacks, while the right ones depict the AUC scores obtained by online attacks. For online attacks, half of reference models are OUT and half are IN (trained per each sample). In LiRA (Carlini et al., 2022) and RMIA, we use 18 augmented queries and for Attack-R and Attack-P (Ye et al., 2022), we use 1 query. For RMIA, we use $\gamma = 2$.

### A.3 PERFORMANCE OF ATTACKS UNDER DATA DISTRIBUTION SHIFT

Table 5 compares the result of attacks when the target models are trained on different datasets than the reference models. More specifically, the target models are trained on CIFAR-10, while the reference models are trained on CINIC-10. We use images with common class labels between two datasets. Compared with other two attacks, RMIA always obtains a higher AUC (e.g. by up to 25% in comparison with LiRA) and a better TPR at low FPRs.

| # Ref. Models | Attack | AUC | TPR@FPR 0.01% | TPR@FPR 0.0% |
|---|---|---|---|---|
| 1 | Attack-R, Ye et al. (2022) | $61.41 \pm 0.23$ | 0.02 | 0.01 |
| | LiRA (Offline), Carlini et al. (2022) | $54.48 \pm 0.21$ | 0.06 | 0.01 |
| | **RMIA (Offline)** | $\mathbf{64.84 \pm 0.25}$ | **0.06** | **0.02** |
| 2 | Attack-R, Ye et al. (2022) | $61.45 \pm 0.30$ | 0.03 | 0.01 |
| | LiRA (Offline), Carlini et al. (2022) | $52.58 \pm 0.18$ | 0.01 | 0.00 |
| | **RMIA (Offline)** | $\mathbf{66.05 \pm 0.29}$ | **0.13** | **0.04** |
| 4 | Attack-R, Ye et al. (2022) | $61.54 \pm 0.30$ | 0.04 | 0.02 |
| | LiRA (Offline), Carlini et al. (2022) | $55.15 \pm 0.25$ | 0.03 | 0.01 |
| | **RMIA (Offline)** | $\mathbf{66.78 \pm 0.32}$ | **0.22** | **0.09** |

Table 5: Performance of offline attacks when different datasets are used for training target and reference models. Specifically, the target models are trained on CIFAR-10, while the reference models are trained on CINIC-10. Results are averaged over 10 random target models.

### A.4 PERFORMANCE OF ATTACKS ACROSS VARIED NEURAL NETWORK ARCHITECTURES

Figure 7 illustrates the performance of attacks when models are trained with different architectures, including CNN and Wide ResNet (WRN) of various sizes, on CIFAR-10. In this scenario, both target and reference models share the same architecture. RMIA consistently outperforms other attacks across all architectures (e.g. 7.5%-16.8% higher AUC compared with LiRA). Using network architectures with more parameters can lead to increased leakage, as shown in (Carlini et al., 2019).

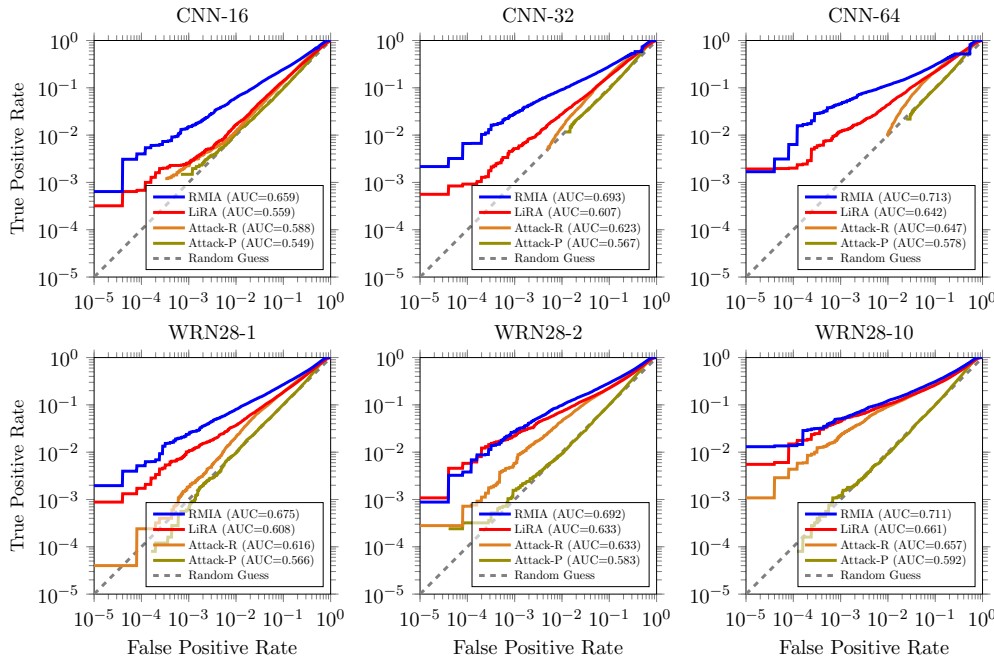

Figure 7: The ROC of attacks using different neural network architectures for training models on CIFAR-10. Here, both target and reference models have the same architecture. We use 2 reference models (1 IN, 1 OUT ). In LiRA (Carlini et al., 2022) and RMIA, we use 18 augmented queries.

Figure 8 presents the performance of attacks when different architectures are used to train reference models, while keeping the target model fixed as WRN28-2. So, the target and reference models may have different architectures. The optimal performance for both RMIA and LiRA is observed when both target and reference models share similar architectures. However, RMIA outperforms other attacks again, and notably, the performance gap widens under architecture shifts.

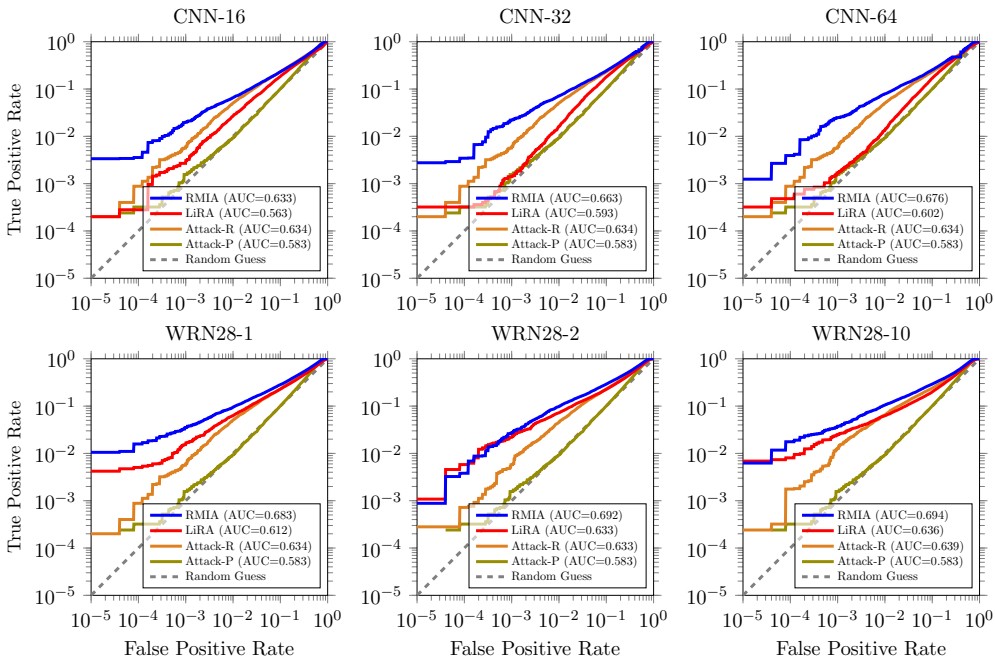

Figure 8: The ROC of attacks using different neural network architectures for training reference models on CIFAR-10. The target model is always trained using WRN28-2. We here use 2 reference models (1 IN, 1 OUT) for each target sample. In LiRA (Carlini et al., 2022) and RMIA, we use 18 augmented queries.

## A.5 ATTACKING DP-SGD

Differential Privacy (DP) serves as a primary defense mechanism against privacy attacks, including membership inference on machine learning models. It establishes an upper bound on the success of any membership inference attack. Similar to the setting used in (Carlini et al., 2022), we assess the impact of DP-SGD (Abadi et al., 2016) on the performance of various attacks, exploring three combinations of DP-SGD's noise multiplier and clipping norm parameters across different privacy budgets: 1) for $\epsilon = \infty$, we set the noise multiplier $\sigma = 0.0$ and clipping gradient $C = 10$ (resulting in a test accuracy of 80.53%), 2) for $\epsilon = 6300$, we use $\sigma = 0.2$ and $C = 5$ (resulting in a test accuracy of 73.59%), and 3) for $\epsilon = 26$, we opt $\sigma = 0.8$ and $C = 1$ (yielding a test accuracy of 49.89%). As noted by (Carlini et al., 2022), well-trained models with DP-SGD exhibit membership inference performance close to random guessing. To enhance model memorization, we deliberately flip the label of a small portion (around 2%) of samples in the population before training. Finally, a balanced set of relabeled member and non-member samples is used to evaluate the attacks.

Figure 9 depicts the ROC of attacks when DP-SGD is used to train CNN-32 models on the CIFAR-10 dataset. For each target sample, we employ 4 reference models (2 IN and 2 OUT). Across all three settings, RMIA outperforms the other three attacks in terms of the obtained AUC. Specifically, in the more relaxed setting (with an infinite budget, zero noise multiplier, and a large clipping gradient), it yields around 3% higher AUC and also a higher TPR at zero FPR than its closest rival. In the other two stringent settings, our attack further enhances its superiority over other attacks concerning AUC (e.g. by at least 6.5% higher AUC than LiRA), although the performance of all attacks tends to converge to random guessing. DP-SGD is shown to be an effective method to mitigate membership inference attacks but comes at the cost of significantly degrading the accuracy of the model.

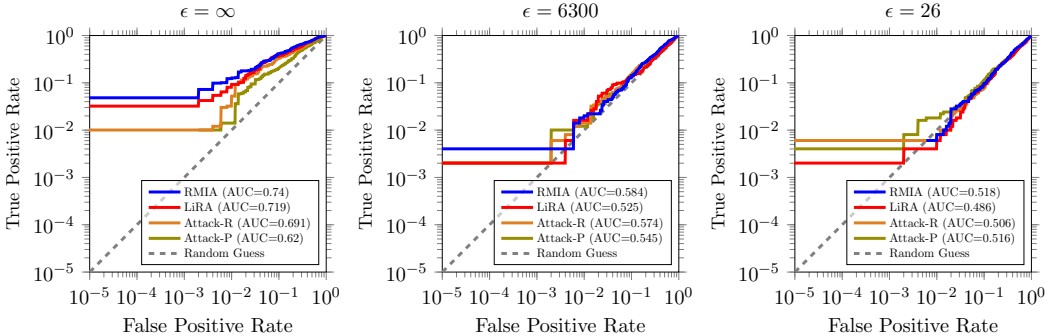

Figure 9: The ROC of attacks using DP-SGD for training CNN-32 models on CIFAR-10. We deliberately flip the label of 2% of samples to enhance model memorization. Three distinct privacy budgets, each accompanied by specific settings for DP-SGD parameters, are used: $\epsilon = \infty$ (left plot), $\epsilon = 6300$ (middle plot), and $\epsilon = 26$ (right plot). Here, we use 4 reference models (2 IN, 2 OUT), for each target sample. In LiRA (Carlini et al., 2022) and RMIA, we use 18 augmented queries.

### A.6 PERFORMANCE OF ATTACKS ON MODELS TRAINED WITH OTHER ML ALGORITHMS

While the majority of contemporary machine learning (ML) models rely on neural networks, understanding how attacks generalize in the presence of other machine learning algorithms is intriguing. Although it is beyond the scope of this paper to comprehensively analyze attacks across a wide range of ML algorithms on various datasets, we conduct a simple experiment to study their impact by training models with a Gradient Boosting Decision Tree (GBDT) algorithm. Figure 10 illustrates the ROC of attacks when GBDT (with three different max depths) is employed to train models on our non-image dataset, i.e. Purchase-100. The hyper-parameters of GBDT are set as n_estimators=250, lr=0.1 and subsample=0.2, yielding a test accuracy of around 53%. For this experiment, we use two reference models (1 IN, 1 OUT) for each target sample. Since the output of GBDT is the prediction probability (not logit), we use this probability as the input signal for all attacks. To compute the rescaled-logit signal for LiRA, we use $\log(\frac{p}{1-p})$, where $p$ is the output probability. The TPR obtained by our attack consistently outperforms all other attacks, particularly by an order of magnitude at zero FPR.

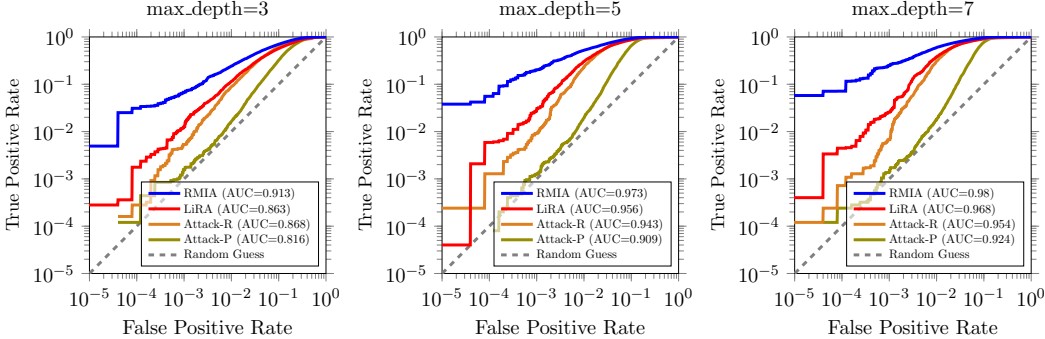

Figure 10: The ROC of attacks on models trained with Gradient Boosted Decision Tree (GBDT) on the Purchase-100 dataset. Three different values of max depth parameter are used in GBDT. The test accuracy of models is around 53%. We use 2 reference models (1 IN, 1 OUT) for each target sample. In LiRA (Carlini et al., 2022), we use the output probability $p$ to compute the rescaled-logit signal as $\log(\frac{p}{1-p})$. In other attacks, we use the output probability as the input signal of attacks. Here, we do not use augmented queries.

## A.7 Performance of Attacks on Typical/Atypical Test Samples

Figure 11 illustrates the performance of attacks using typical and atypical test samples on CIFAR-10 models. Typical (interior) samples are characterized by the highest model confidence across all reference models, while atypical (boundary) samples exhibit the lowest confidence. In each experiment, the number of member and non-member test samples is balanced. The left plots in this figure depict results with only 1 (OUT) reference model, whereas the right plots showcase outcomes with 254 reference models (127 IN, 127 OUT) for each target sample.

In all four scenarios, RMIA outperforms other attacks in terms of the achieved AUC. When using numerous reference models, RMIA's performance is marginally superior to LiRA, particularly with atypical test samples. However, with only 1 reference model, our attack demonstrates a notable improvement in AUC compared to LiRA (approximately 35% and 7% for atypical and typical samples, respectively). Attacking models with respect to typical samples proves to be a much more challenging task, because these samples are less memorized by the target model, and there is a subtle distinction in their MIA scores whether they are in the training set or not.

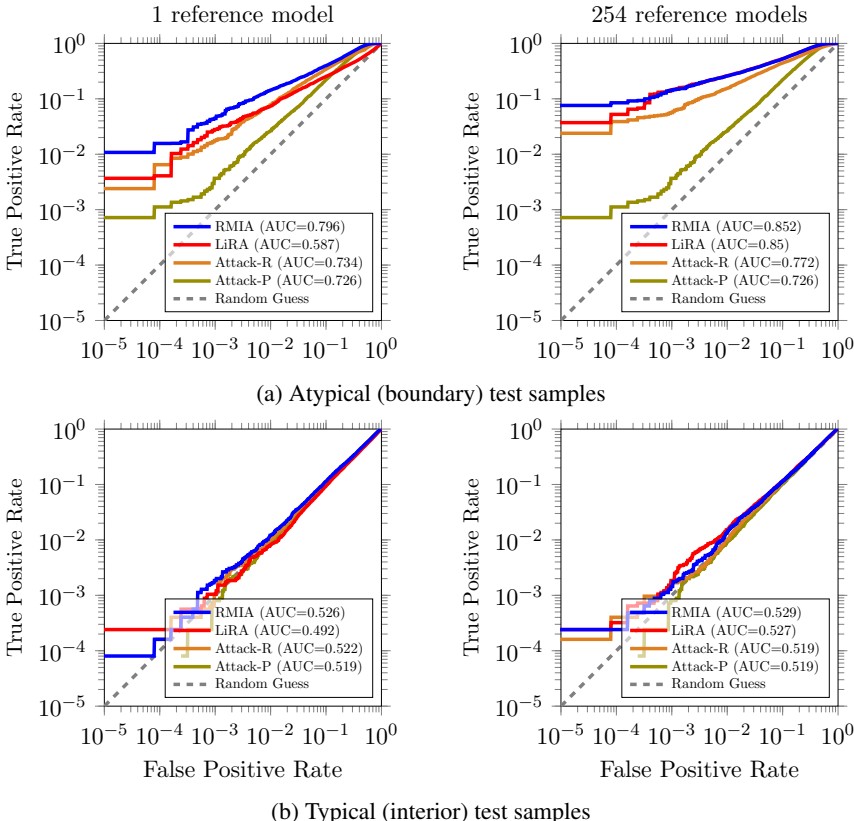

Figure 11: The ROC of attacks using typical and atypical test samples on CIFAR-10 models. Typical samples refer to those with the highest model confidence across all reference models, while atypical samples are those with the lowest confidence. The left plots showcase the results with 1 (OUT) reference model, whereas the right ones display results with 254 models (127 IN, 127 OUT) for each target sample. In LiRA (Carlini et al., 2022) and RMIA, we use 18 augmented queries.

## A.8 The Impact of Number of Reference Records

Our attack relies on evaluating the likelihood ratio for generating the target model by an input query $x$ and other non-member $z$ samples, as formulated in equation 2. Note that we choose non-member samples randomly from the population data. Table 6 shows how the result of RMIA changes, as we use different number of reference samples. The AUC increases when we grow the number of $z$ samples, but it is noteworthy that using 2500 reference samples, equivalent to 10% of the size of the

models' training set, yields results comparable to those obtained with the entire reference sample set. Moreover, the TPR at low FPRs is still high even when we consider just 250 reference samples. In the default setting, we use all non-members to a given target model as the set of $z$ samples.

| # z samples | AUC | TPR@FPR | |
| --- | --- | --- | --- |
| | | 0.01% | 0.0% |
| 25 samples | $58.71 \pm 0.26$ | 1.32 | 1.15 |
| 250 samples | $64.88 \pm 0.24$ | 2.23 | 1.65 |
| 1250 samples | $67.57 \pm 0.30$ | 2.22 | 1.73 |
| 2500 samples | $68.25 \pm 0.27$ | 2.26 | 1.72 |
| 6250 samples | $68.78 \pm 0.31$ | 2.28 | 1.75 |
| 12500 samples | $69.03 \pm 0.33$ | 2.28 | 1.77 |
| 25000 samples | $69.15 \pm 0.35$ | 2.26 | 1.80 |

Table 6: Performance of RMIA using different number of random $z$ samples for each target sample. The maximum number equals using all samples in the population, except those used for training the target model. The number of reference models, trained with CIFAR-10, is 254. We use no augmented queries, and $\gamma = 2$.

### A.9 RELATION BETWEEN $\gamma$ AND $\beta$

We here investigate the impact of $\gamma$ and $\beta$ selection on the efficacy of our attack. In Figure 12, we illustrate the sensitivity of our attack, measured in terms of AUC and FPR@TPR, to changes in the value of $\gamma$. We do not show the result for $\gamma < 1$, because it implies that a target sample $x$ is allowed to have a lower chance of being member than reference samples to pass our pairwise likelihood ratio test, causing lots of non-members to be wrongly inferred as member. With increasing $\gamma$, both AUC and FPR@TPR remain relatively stable (except for instances with considerably high values of $\gamma$). This behavior is elucidated in Figure 13 which demonstrates the best value of $\beta$ to optimize TPR at given FPRs for different $\gamma$ values. As $\gamma$ increases, the need arises to decrease the value of $\beta$ in the hypothesis test equation (equation 1) to strike a balance between the power and error of the attack. By appropriately adjusting the value of $\beta$, we achieve a roughly same result across different $\gamma$ values. In extreme cases where a very high $\gamma$ is employed, the detection of $\gamma$-dominated reference samples between a limited set of $z$ records becomes exceedingly challenging. Consequently, a drop in both metrics is observed. We consistently note the same trend in results across all our datasets and with varying numbers of reference models. Throughout our experiments, we set $\gamma$ to 2, as it yields a slightly higher TPR@FPR. With $\gamma = 2$, the best value of $\beta$ to maximize TPR at 0.01% and 0% FPR is about 0.21 and 0.27, respectively.

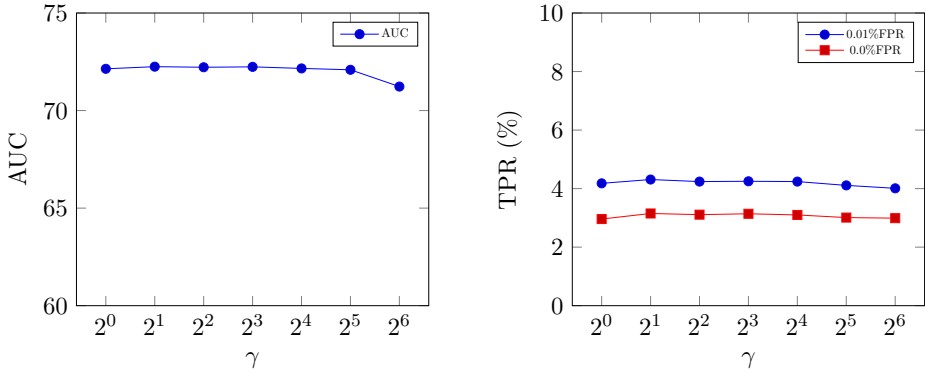

Figure 12: The performance sensitivity of our attack (RMIA) with respect to $\gamma$ parameter. Here, we use 254 reference models trained on CIFAR-10. The left plot shows the AUC obtained when using different $\gamma$ values, while the right plot demonstrates the TPR at 0.01% and 0% FPR values versus $\gamma$ (corresponding to blue and red lines, respectively). We use 18 augmented queries. Results are averaged over 10 random target models.

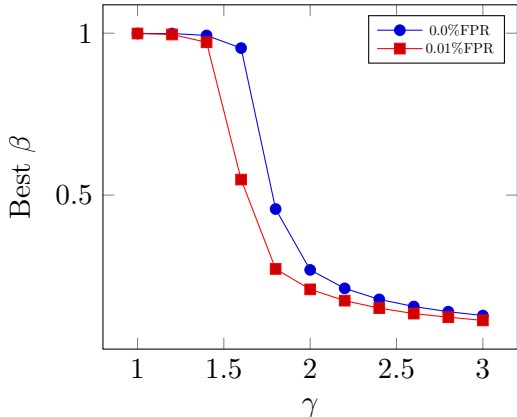

Figure 13: The best value of $\beta$ that maximizes TPR at FPR=0.01% (red line) and FPR=0% (blue line) for each given $\gamma$ value. The number of reference models, trained with CIFAR-10, is 254 and we use no augmented queries.

Figure 14 illustrates the impact of selecting $\gamma$ and $\beta$ on the TPR and FPR of RMIA. Plots in this figure are obtained by using three different $\gamma$ values, i.e. $\gamma$ =0.9, 1 and 2, respectively (from left to right). Each plot shows the value of TPR and FPR when $\beta$ changes from 0 to 1 under the given $\gamma$ value. Both TPR and FPR always approach zero, as we increase $\beta$ to 1 and the reason is that it is unlikely that a target sample can dominate most of reference records. But, FPR decreases more rapidly than TPR under all $\gamma$ values which allows us to always have a higher power gain than error. From the figure, we realize that if our goal is to maximize TPR at a very low FPR, we have to set $\gamma$ to a value larger than 1, because with $\gamma = 0.9$ or 1, we cannot reduce FPR to zero unless we use $\beta = 1$ which gives us a zero TPR as well. In a broader sense, when $\gamma$ is set to 1, the value of $\beta$ shows a strong correlation with the value of 1 - FPR.

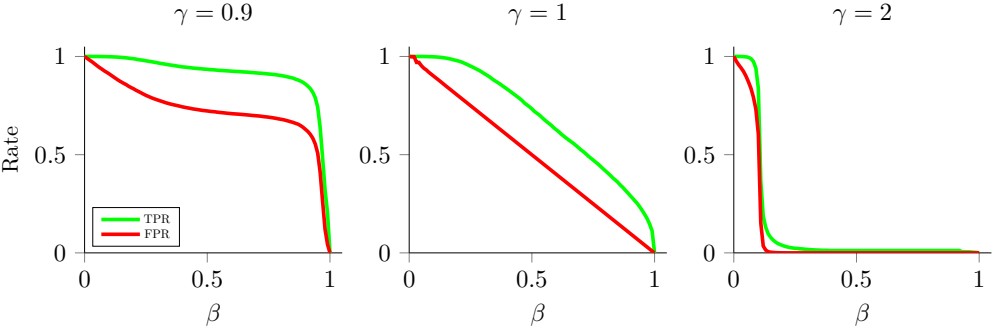

Figure 14: TPR and FPR achieved by RMIA for different values of $\gamma$ and $\beta$. From left to right, the plots correspond to the result obtained for $\gamma = 0.9$, 1, and 2, respectively. The number of reference models, trained with CIFAR-10, is 254 and we use no augmented queries.

## A.10 COMPUTING THE PROBABILITIES FOR THE LIKELIHOOD RATIO IN EQUATION 4

### A.10.1 COMPUTING $\Pr(x|\theta)$

Let $\theta$ be a classification (neural network) model that maps each input data $x$ to a probability distribution across $k$ classes. Assume data point $x$ is in class $y$. Let $c(x) = \langle c_1, \cdots, c_k \rangle$ be the output vector (logits) of the neural network for input $x$, before applying the final normalization. We denote the normalized prediction probability of class $y$ for the input $x$ as $f_\theta(x)_y$. For the Softmax function, the probability is given by

$$f_\theta(x)_y = \frac{e^{\frac{c_y}{T}}}{\sum_{i=1}^{k} e^{\frac{c_i}{T}}},$$

where $T$ is a temperature constant. We can use this to estimate $\Pr(x|\theta)$. However, there are many alternatives to Softmax probability proposed in the literature to improve the accuracy of estimating $\Pr(x|\theta)$, using the Taylor expansion of the exponential function, and using heuristics to refine the relation between the probabilities across different classes (De Brebisson & Vincent, 2016; Liu et al., 2016; Liang et al., 2017; Banerjee et al., 2021). In this paper, we use the following to compute $\Pr(x|\theta)$ for image datasets:

$$f_\theta(x)_y = \frac{apx(c_y - m)}{apx(c_y - m) + \sum_{i \neq y} apx(c_i)}, \tag{7}$$

where $apx(a) = \sum_{i=0}^{n} \frac{a^i}{i!}$ is the $n$th order Taylor approximation of $e^a$, and $m$ is a hyperparameter that controls the separation between probability of different classes. We compare our attack results in case of using different prediction likelihood functions.

### A.10.2 IMPACT OF PREDICTION LIKELIHOOD FUNCTION

We here study the choice of the prediction likelihood function (signal) on the outcome of RMIA. There are several alternatives to the Softmax function for computing the prediction likelihood of models, including Taylor-Softmax (De Brebisson & Vincent, 2016), Soft-Margin Softmax (SM-Softmax) (Liang et al., 2017), and their combination (SM-Taylor-Softmax) (Banerjee et al., 2021). In Table 7, we compare the result of RMIA obtained with four different signals. Based on our empirical results, we set the soft-margin $m$ and the order $n$ in Taylor-based functions to 0.6 and 4, respectively. The temperature ($T$) is set to 2 for CIFAR-10 models. SM-Taylor-Softmax obtains the highest AUC, while also achieving a rather better TPR at low FPRs. Hence, we use SM-Taylor-Softmax function, as formulated in equation 7, for our image datasets. Figure 15 presents the performance sensitivity of our attack in terms of AUC concerning three hyper-parameters in this function: order $n$, soft-margin $m$, and temperature $T$. The AUC of RMIA appears to be robust to variations in $n$, $m$ and $T$. For $n \geq 3$, the results are consistent, but employing lower orders leads to a poor Taylor-based approximation for Softmax, as reported in other works (Banerjee et al., 2021).

| Likelihood Function | AUC | TPR@FPR | |
| --- | --- | --- | --- |
| | | 0.01% | 0.0% |
| Softmax | $68.96 \pm 0.31$ | 2.25 | 1.69 |
| SM-Softmax | $69.02 \pm 0.34$ | **2.27** | 1.67 |
| Taylor-Softmax | $68.57 \pm 0.29$ | 2.06 | 1.43 |
| SM-Taylor-Softmax | $\mathbf{69.15 \pm 0.35}$ | 2.26 | **1.80** |

Table 7: Performance of RMIA obtained with various prediction likelihood functions. The number of reference models, trained with CIFAR-10, is 254, we use no augmented queries, and $\gamma = 2$. The soft-margin $m$ and the order $n$ in Taylor-based functions are 0.6 and 4, respectively.

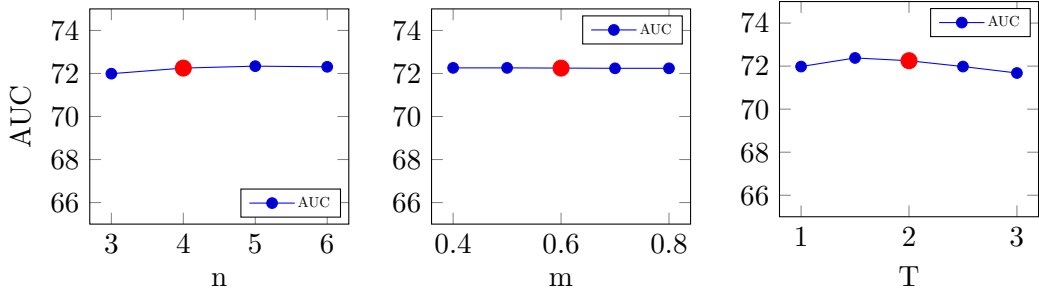

Figure 15: The AUC of our attack (RMIA) obtained by using different values of $n$ (order in Taylor function), $m$ (soft-margin) and $T$ (temperature) in SM-Taylor-Softmax function. When modifying one parameter, we hold the values of the other two parameters constant at their optimal values. Here, we use 254 reference models trained on CIFAR-10 and 18 augmented queries. Results are averaged over 10 target models. The red points indicate the default values used in our experiments.

A.10.3  COMPUTING $\Pr(x)$

Recall that $\Pr(x)$ is the normalizing constant for the Bayes rule in computing our LR. See equation 5. We compute it as the empirical average of $\Pr(x|\theta')$ on reference models $\theta'$ trained on random datasets $D$ drawn from the population distribution $\pi$. This is then used in pairwise likelihood ratio equation 4.

In order to compute $\Pr(x)$, we need to train reference models $\theta'$. Note that the reference models must be sampled in an unbiased way with respect to whether $x$ is part of their training data. This is because the summation in equation 5 is over all $\theta'$, which can be partitioned to the set of models trained on $x$ (IN models), and the set of models that are not trained on x (OUT models). Let $\bar{x}$ denote that a training set does not include $x$. Let $\theta'_x$ denote an IN model trained on dataset $D_x$ ($x \in D_x$) and $\theta'_{\bar{x}}$ be an OUT model trained on dataset $D_{\bar{x}}$ ($x \notin D_{\bar{x}}$). Then, from equation 5, we have:

$$\Pr(x) = \sum_{\theta',D} \Pr(x|\theta') \Pr(\theta'|D) \Pr(D) = \sum_{\theta'_x,D_x} \Pr(x|\theta'_x) \Pr(\theta'_x|D_x) \Pr(D_x)$$
$$+ \sum_{\theta'_{\bar{x}},D_{\bar{x}}} \Pr(x|\theta'_{\bar{x}}) \Pr(\theta'|D_{\bar{x}}) \Pr(D_{\bar{x}}) \tag{8}$$

The two sums on the right-hand side of the above equation can be computed empirically using sampling methods. Instead of integrating over all possible datasets and models, we sample datasets $D_x$ and $D_{\bar{x}}$ and models $\theta'_x$ and $\theta'_{\bar{x}}$, and compute the empirical average of $\Pr(x|\theta'_x)$ and $\Pr(x|\theta'_{\bar{x}})$ given the sampled models. We sample $D_{\bar{x}}$ from the probability distribution $\Pr(D_{\bar{x}})$, which is the underlying data distribution $\pi$. For sampling $D_x$, we sample a dataset from $\pi$ and add $x$ to the dataset. We sample $\theta'_x$ and $\theta'_{\bar{x}}$ by training models on $D_x$ and $D_{\bar{x}}$, respectively.

In the online setting for MIA, we can empirically estimate $\Pr(x)$ by computing the average $\Pr(x|\theta')$ over 50% IN models and 50% OUT models (using $2k$ models), i.e.:

$$\Pr(x) \approx \frac{1}{2} \Big( \underbrace{\frac{1}{k} \sum_{\theta'_x} \Pr(x|\theta'_x)}_{\Pr(x)_{IN}} + \underbrace{\frac{1}{k} \sum_{\theta'_{\bar{x}}} \Pr(x|\theta'_{\bar{x}})}_{\Pr(x)_{OUT}} \Big) \tag{9}$$

However, in the offline setting, we do not have access to IN models. Thus, we need to exclusively use $\Pr(x|\theta'_{\bar{x}})$. As a simple heuristic to have a less biased estimate of $\Pr(x)$ without having access to IN models, we perform an offline pre-computation to approximate the shift of probability between OUT and IN models when we fine-tune a reference model with randomly drawn samples from the population. Essentially we approximate the sensitivity of models (the gap between probability of points when they are non-member and when they become members). See Figure 1 in (Zhang et al., 2021) for such computation.

We use our existing reference models to compute the rate at which $\Pr(x)$ changes between reference models that inclue $x$ versus the others. If we have very few (e.g., 1 reference model), we compute how much the probability of population data change if we fine-tune the model on them. We approximate the gap with a linear function, so we obtain $\Pr(x)_{IN} = a.\Pr(x)_{OUT} + b$, and finally can obtain $\Pr(x) = (\Pr(x)_{IN} + \Pr(x)_{OUT})/2$. Given that both $\Pr(x)_{IN}$ and $\Pr(x)_{OUT}$ fall within the range of 0 to 1, it follows that $a + b = 1$. Consequently, we can simplify the linear function as $\Pr(x)_{IN} = a.(\Pr(x)_{OUT} - 1) + 1$. In Figure 16, we present the AUC obtained by our offline RMIA on CIFAR-10 models, varying the value of $a$ from 0 to 1. As $a$ approaches 1, there is a degradation in AUC (by up to 5.5%), because we heavily rely on $\Pr(x)_{OUT}$ to approximate $\Pr(x)_{IN}$. Lower values of $a$ result in an enhancement of $\Pr(x)_{OUT}$ to approximate $\Pr(x)_{IN}$, leading to improved performance. The AUC appears to be more robust against lower values of $a$, particularly those below 0.5. Notably, even with $a = 0$, where $\Pr(x) = (1 + \Pr(x)_{OUT})/2$, a considerable improvement in results is observed. In this case, we are alleviating the influence of very low $\Pr(x)_{OUT}$ for atypical/hard samples.

In our experiments, we empirically derived the following values for the datasets used in this paper: $a = 0.33$ for CIFAR-10 and CINIC-10, $a = 0.6$ for CIFAR-100, and $a = 0.2$ for Purchase-100 models.

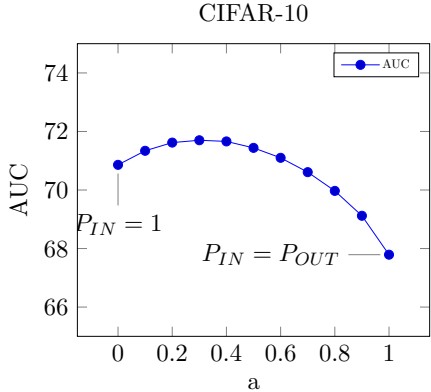

Figure 16: The AUC of offline RMIA obtained by using different values of $a$ in the linear approximation function. Here, we use 127 OUT reference models trained on CIFAR-10. We use 18 augmented queries. Results are averaged over 10 random target models.

### A.11 MIA SCORE COMPARISON BETWEEN ATTACKS

To better understand the difference between the performance of our attack with others' (specially when focusing on low FPRs), we compare the MIA score of member and non-member samples obtained in RMIA and other attacks. Figure 17 displays RMIA scores versus LiRA scores in two scenarios: one using only 1 reference model (shown on the left) and another using 4 reference models (shown on the right). With just 1 reference model, RMIA provides clearer differentiation between numerous member and non-member samples, as it assigns distinct MIA scores in the $[0, 1]$ range, separating many members on the right side and non-members on the left side. In contrast, LiRA scores are more concentrated towards the upper side of the plot, lacking a distinct separation between member and non-member scores. When employing more reference models, we observe a higher degree of correlation between the scores of the two attacks.

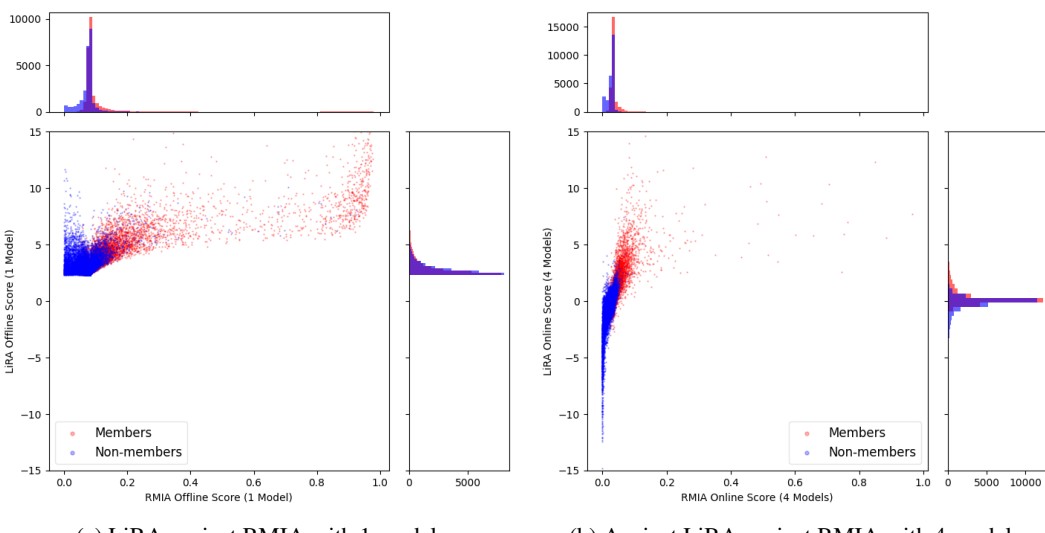

(a) LiRA against RMIA with 1 model          (b) Against LiRA against RMIA with 4 models

Figure 17: MIA score comparison between RMIA and LiRA (Carlini et al., 2022) over all member and non-member samples of a random target model. The left plot is obtained when only 1 (OUT) reference model is used, while the right plot uses 4 reference models per sample (2 IN and 2 OUT). In both plots, the x-axis represents RMIA scores, while the y-axis depicts LiRA scores.

Similarly, Figure 18 shows RMIA scores compared to Attack-R scores in the same two scenarios: using only 1 reference model (shown on the left) and using 4 reference models (shown on the right). For both scenarios, RMIA can better separate members from non-members via assigning distinct MIA scores to them (member scores apparently tend to be larger than non-member scores for lots of samples). In contrast, there is no such clear distinction in Attack-R .

We finally show RMIA scores versus Attack-P scores in Figure 19. In this experiment, we only use 1 (OUT) reference model for RMIA, because Attack-P does not work with reference models. As opposed to RMIA, Attack-P clearly fails to provide a good separation between member and non-member scores.

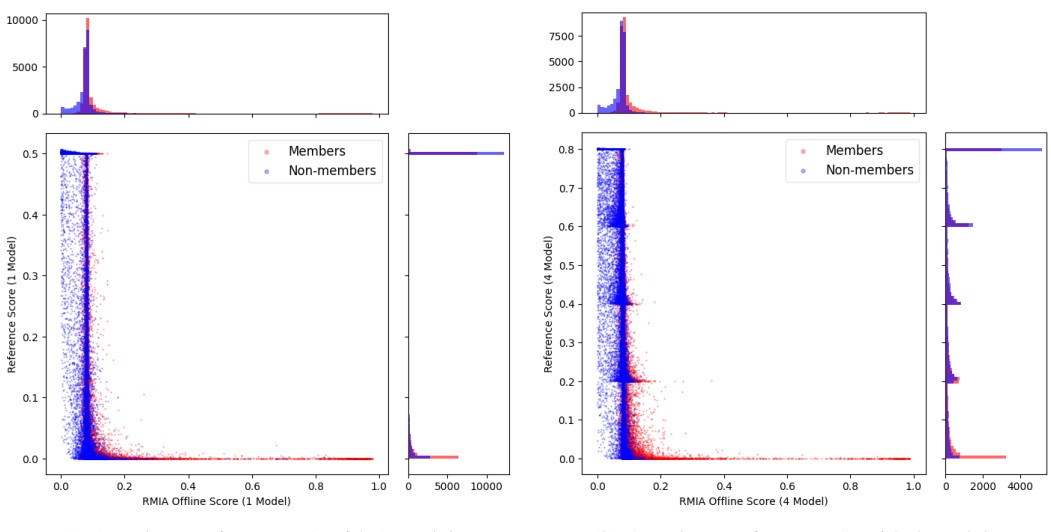

(a) Attack-R against RMIA with 1 model        (b) Attack-R against RMIA with 4 models

Figure 18: MIA score comparison between RMIA and Attack-R (Ye et al., 2022) over all member and non-member samples of a random target model. The left plot is obtained when only 1 (OUT) reference model is used, while the right plot uses 4 (OUT) reference models per sample. In both plots, the x-axis represents RMIA scores, while the y-axis depicts Attack-R scores.

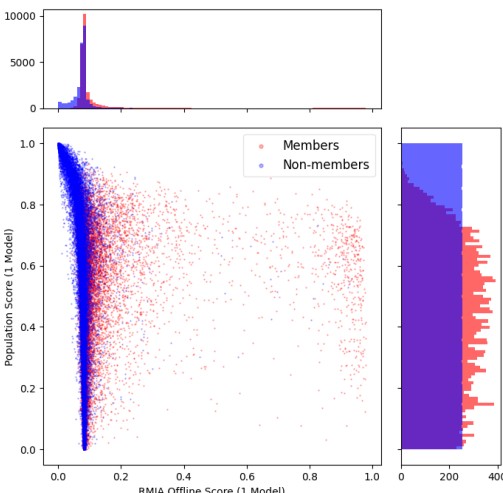

Figure 19: MIA score comparison between RMIA and Attack-P (Ye et al., 2022) over all member and non-member samples of a random target model. The x-axis represents RMIA scores, while the y-axis depicts Attack-P scores. RMIA results are computed using only 1 (OUT) reference model.

A.12   DIRECT COMPUTATION OF LIKELIHOOD RATIO IN EQUATION 2

In Section 2, we introduced two separate approaches for computing the fundamental likelihood ratio in equation 2: I) a Bayesian method, as shown by equation 4, and II) a direct method, expressed in equation 6. While our empirical results are primarily derived using the Bayesian method, we anticipate that the outcomes of these two approaches will converge closely when a sufficient number of reference models is employed, although their performance may vary with a limited number of models. In this section, we attempt to assess these two methods and possibly invalidate our initial anticipation.

The direct approach involves employing Gaussian modeling over logits. Our attack is then simplified to the estimation of the mean and variance for the logits of samples $x$ and $z$ in two distinct distributions of reference models; One distribution comprises models trained with $x$ in their training set but not $z$ (denoted by $\theta'_{x,\bar{z}}$), while the other comprises models trained with $z$ in their training set but not $x$ (denoted by $\theta'_{\bar{x},z}$). Let $\mu_{x,\bar{z}}(x)$ and $\sigma_{x,\bar{z}}(x)$ be the mean and variance of $f_{\theta'}(x)$, i.e. the logit of sample $x$, in the distribution of $\theta'_{x,\bar{z}}$ models, respectively (a similar notation can be defined for $\theta'_{\bar{x},z}$ models). We can approximate the likelihood ratio in equation 6 as follows:

$$
\begin{aligned}
\mathrm{LR}_\theta(x, z) &= \frac{\Pr(f_\theta(x), f_\theta(z)|x)}{\Pr(f_\theta(x), f_\theta(z)|z)} \\
&\approx \frac{\Pr(f_\theta(x)|\mathcal{N}(\mu_{x,\bar{z}}(x), \sigma^2_{x,\bar{z}}(x))) . \Pr(f_\theta(z)|\mathcal{N}(\mu_{x,\bar{z}}(z), \sigma^2_{x,\bar{z}}(z)))}{\Pr(f_\theta(x)|\mathcal{N}(\mu_{\bar{x},z}(x), \sigma^2_{\bar{x},z}(x))) . \Pr(f_\theta(z)|\mathcal{N}(\mu_{\bar{x},z}(z), \sigma^2_{\bar{x},z}(z)))}
\end{aligned}
\tag{10}
$$

where $f_\theta(x)$ represents the output (logits) of the target model $\theta$ on sample $x$. Although the direct method may appear to be more straightforward and accurate, it comes with a significantly higher computational cost, since we must train online reference models, i.e. $\theta'_{x,\bar{z}}$, to compute the probabilities in the above relation.

We now demonstrate the performance of our RMIA when the likelihood ratio is computed using equation 4 (hereafter, called RMIA-Bayes), in comparison to utilizing the aforementioned likelihood ratio in equation 10 (referred to as RMIA-direct). Figure 20 compares the ROCs achieved by the two methods when 64 reference models are employed for each sample pair $x$ and $z$ to estimate the probabilities. We show the results obtained from models trained with different datasets. In this case, RMIA-direct slightly outperforms in terms of both AUC and TPR at low FPRs, yet RMIA-Bayes closely matches its pace, even at low FPR values. On the other hand, Figure 21 presents the scenario where 4 reference models are used for each sample pair. When utilizing fewer models, RMIA-Bayes exhibits a better performance across all datasets (e.g., 6.6% higher AUC in CIFAR-10). It appears that RMIA-direct struggles to accurately estimate the parameters of Gaussian models with only four models available. Given the substantial processing cost of RMIA-direct, associated with training online models relative to sample pairs, RMIA-Bayes emerges as a more reasonable choice due to its ability to operate with a reduced number of models trained in an offline context.

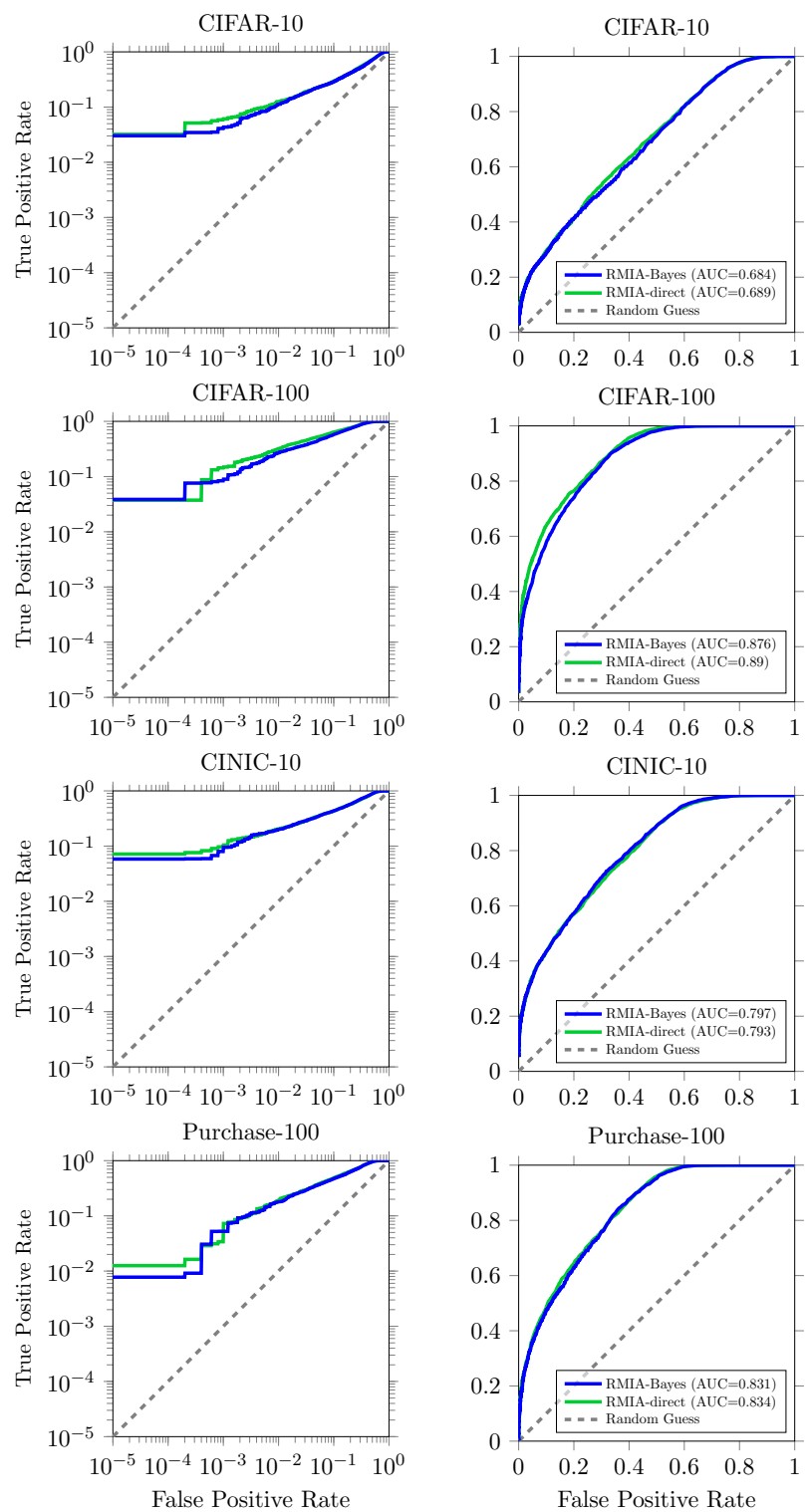

Figure 20: The ROC of RMIA when two different methods, i.e. RMIA-direct and RMIA-Bayes, are used to approximate the likelihood ratio in equation 2 (ROCs are shown in both log and normal scales). Here, 64 models are trained per each sample pair $x$ and $z$ on different datasets. We use no augmented queries. We also set $\gamma$ to 2.

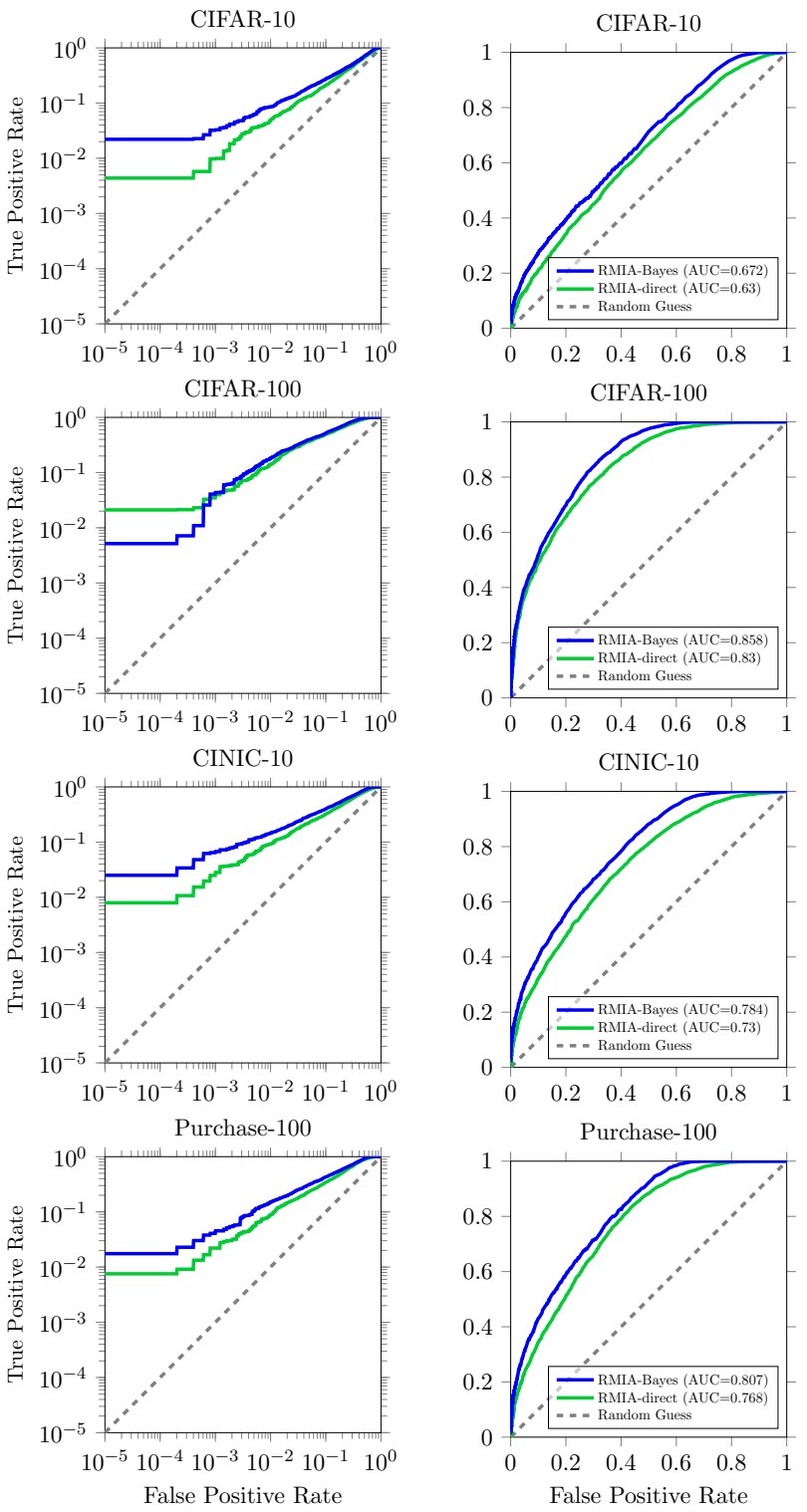

Figure 21: The ROC of RMIA when two different methods, i.e. RMIA-direct and RMIA-Bayes, are used to approximate the likelihood ratio in equation 2 (ROCs are shown in both log and normal scales). Here, 4 models are trained per each sample pair $x$ and $z$ on different datasets. We use no augmented queries. We also set $\gamma$ to 2.

## A.13 PREDICTION VARIANCE FOR POPULATION SAMPLES ACROSS REFERENCE MODELS

In Section 2.2, we introduced using reference models for computing $\Pr(x)$ and $\Pr(z)$ in the likelihood ratio. Reference models are trained on random samples from the population. Here, we assess the distributions of the average probability of non-member samples across various models trained on our 4 datasets, to analyze the variance that the difference in these models can create in the computation of $\Pr(x)$ and $\Pr(z)$. Specifically, for each dataset examined, we train 254 models, and compute the average prediction probability of their test datasets on each of these models. Figure 22 illustrates the distribution of these average probabilities, aggregated across all models. Here, the x-axis represents the distribution of per-model average prediction probability of test data, and the y-axis indicates the number of models. Notably, as can be observed in the figure, the average probabilities cluster closely around the mean, and the standard deviation from the mean is minimal, consistently staying below $0.003$ across all datasets examined. This shows that the computation of $\Pr(x)$ and $\Pr(z)$ should not be too sensitive to adding or removing a reference model trained on random samples from the population.

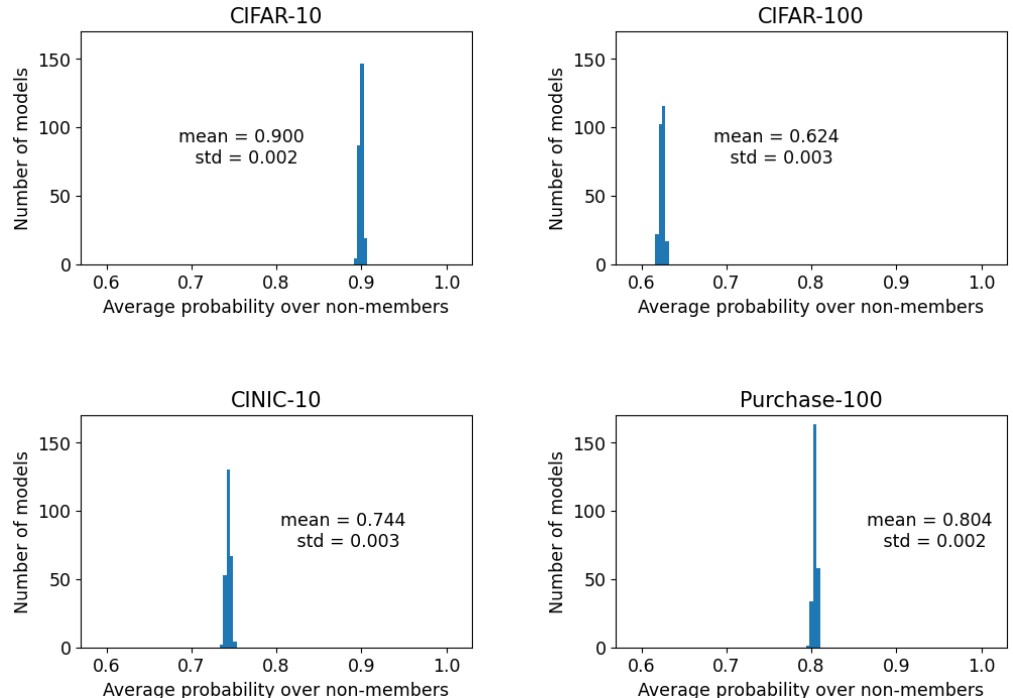

Figure 22: Distributions of average prediction probability for non-member samples across trained models. For each dataset, we train 254 models with half of the dataset randomly selected as the training set in each instance. Subsequent to the training of each model, we calculate the average prediction probability for non-member samples corresponding to that particular model. Ultimately, we illustrate the distribution of these average probabilities, accumulated across all models. The x-axis represents the average prediction probability of non-member samples, as computed within each model, while the y-axis denotes the number of models.

# B  RELATED WORK

Neural networks, particularly when trained with privacy-sensitive datasets, have been proven to be susceptible to leaking information about their training data. A variety of attacks have been designed to gauge the degree of leakage and the subsequent privacy risk associated with training data. For instance, *data extraction attacks* attempt to recreate individual samples used in training the model (Carlini et al., 2021), whereas *model inversion attacks* focus on extracting aggregate information about specific sub-classes instead of individual samples (Fredrikson et al., 2015). In contrast, *property inference attacks* derive non-trivial properties of samples in a target model's training dataset (Ganju et al., 2018). This paper, however, is concerned with *membership inference attacks (MIAs)*, which predict whether a particular sample was used in the training process of the model. MIAs, due to their simplicity, are commonly utilized as auditing tools to quantify data leakage in trained models.

MIAs first found their use in the realm of genome data to identify the presence of an individual's genome in a mixed batch of genomes (Homer et al., 2008). Backes et al. (2016) went on to formalize the risk analysis for identifying an individual's genome from aggregate statistics on independent attributes, with this analysis later extended to include data with dependent attributes (Murakonda et al., 2021).

Algorithms featuring differential privacy (DP) are designed to limit the success rate of privacy attacks when distinguishing between two neighboring datasets (Nasr et al., 2021). Some researches, such as Thudi et al. (2022), provide upper limits on the average success of MIAs on general targets. Other studies evaluate the effectiveness of MIAs on machine learning models trained with DP algorithms (Rahman et al., 2018).

Shokri et al. (2017) demonstrated the efficacy of membership inference attacks against machine learning models in a setting where the adversary has query access to the target model. This approach was based on the training of reference models, also known as shadow models, with a dataset drawn from the same distribution as the training data. Subsequent works extended the idea of shadow models to different scenarios, including white-box (Leino & Fredrikson, 2020; Nasr et al., 2019; Sablayrolles et al., 2019) and black-box settings (Song & Mittal, 2021; Hisamoto et al., 2020; Chen et al., 2021), label-only access (Choquette-Choo et al., 2021; Li & Zhang, 2021), and diverse datasets (Salem et al., 2019). However, such methods often require the training of a substantial number of models upon receiving an input query, making them unfeasible due to processing and storage costs, high response times, and the sheer amount of data required to train such a number of models. MIA has been also applied in other machine learning scenarios, such as federated learning (Nasr et al., 2019; Melis et al., 2019; Truex et al., 2019) and multi-exit networks (Li et al., 2022).

Various mechanisms have been proposed to defend against MIAs, although many defense strategies have proven less effective than initially reported (Song & Mittal, 2021). Since the over-fitting issue is an important factor affecting membership leakage, several regularization techniques have been used to defend against membership inference attacks, such as L2 regularization, dropout and label smoothing (Shokri et al., 2017; Salem et al., 2019; Liu et al., 2022b). Some recent works try to mitigate membership inference attacks by reducing the target model's generalization gap (Li et al., 2021; Chen et al., 2022) or self-distilling the training dataset (Tang et al., 2022). Abadi et al. (2016) proposed DP-SGD method which adds differential privacy (Dwork, 2006) to the stochastic gradient descent algorithm. Subsequently, some works concentrated on reducing the privacy cost of DP-SGD through adaptive clipping or adaptive learning rate (Yu et al., 2019; Xu et al., 2020). In addition, there are defense mechanisms, such as AdvReg (Nasr et al., 2018) and MemGuard (Jia et al., 2019), that have been designed to hide the distinctions between the output posteriors of members and non-members.

Recent research has emphasized evaluating attacks by calculating their true positive rate (TPR) at a significantly low false positive rate (FPR) (Carlini et al., 2022; Ye et al., 2022; Liu et al., 2022a; Long et al., 2020; Watson et al., 2022a). For example, Carlini et al. (2022) found that many previous attacks perform poorly under this evaluation paradigm. They then created an effective attack based on a likelihood ratio test between the distribution of models that use the target sample for training (IN models) and models that do not use it (OUT models). Despite the effectiveness of their attack, especially at low FPRs, it necessitates the training of many reference models to achieve high performance. Watson et al. (2022a) constructed a membership inference attack incorporating sample

hardness and using each sample's hardness threshold to calibrate the loss from the target model. Ye et al. (2022) proposed a template for defining various MIA games and a comprehensive hypothesis testing framework to devise potent attacks that utilize reference models to significantly improve the TPR for any given FPR. Liu et al. (2022a) presented an attack which utilizes the membership signals generated during the training of distilled models to effectively enhance the differentiation between members and non-members. The recent paper (Wen et al., 2023) has improved the performance of likelihood test-driven attacks by estimating a variant of the target sample through minimally perturbing the original sample, which minimizes the fitting loss of IN and OUT shadow models. Lately, Leemann et al. (2023) have introduced a novel privacy notion called $f$-MIP, which allows for bounding the trade-off between the power and error of attacks using a function $f$. This is especially applicable when models are trained using gradient updates. The authors demonstrated the use of DP-SGD to achieve this $f$-MIP bound. Additionally, they proposed a cost-effective attack for auditing the privacy leakage of ML models, with the assumption that the adversary has white-box access to gradients of the target model.

In alignment with recent research, we concentrate on the power (TPR) of MIAs at extremely low errors (FPR). We propose a different game and hypothesis test that allows us to devise a new attack that reaps the benefits of various potent attacks. The attack proposed in this paper demonstrates superior overall performance and a higher TPR at zero FPR than the attacks introduced by (Carlini et al., 2022; Ye et al., 2022), especially when only a limited number of reference models are trained. Additionally, it maintains high performance in an offline setting where none of the reference models are trained with the target sample. This characteristic renders our attack suitable for practical scenarios where resources, time, and data are limited.

