# OpenReview forum: "Low-Cost High-Power Membership Inference by Boosting Relativity"
_ICLR.cc/2024/Conference — Submitted to ICLR 2024_

### Official Review · Reviewer_F7FX · 2023-10-20

**Soundness:** 3 good
**Presentation:** 2 fair
**Contribution:** 3 good
**Rating:** 6
**Confidence:** 3

**Summary:**

The paper proposes a new membership inference attack against machine learning models. Numerical experiments on four data sets demonstrate its superior performance over existing attacks. Some heuristic justifications are given to explain the improvement in performance.

**Strengths:**

* The new membership inference attack proposed by this paper consistently outperforms existing attacks, particularly when the number of reference models available is small. This advantage significantly reduces the computation burden of membership inference attacks.

* There are some neat heuristics for explaining the improvement in empirical performance. These heuristics are useful for inspiring new attack methods, and/or theoretical research on privacy attacks (and privacy-preserving machine learning in general).

**Weaknesses:**

1. *Theoretical justification of the method*. Section 2.2 attempts to justify the "test statistic", equation (4), by arguing that it is a good approximation of the "true" likelihood ratio, equation (3). In particular, the paragraph between equations (3) and (4) and the paragraph between equations (4) and (5) makes many assumptions and makes several approximations in succession. It might be easier for readers if these assumptions and approximations are spelled out more directly, preferably in more precise mathematical notations (For a few examples, the "true" quantity and the "estimate" are both referred to as "LR"; the quantity "MIA" is first defined as a probability in equation (5), but the actual attack in the paper is an approximation of the "MIA". )

2. *Dependence on reference records*. While the new attack's robustness to few reference models has been clearly demonstrated, I wonder whether the new attack's dependence on many reference records is replacing one type of constraint (abundance of reference models) with another type of constraint (free access to additional samples from the population). While there are some encouraging results in the appendix, 10% of the entire population appears to be a very large amount in most practical applications (sometimes, it is not known in advance what the "population" is, or how large the "population" might be).

3. *Lack of practical criteria for selecting input parameters $\gamma$ and $\beta$*. There is not much discussion on why $\gamma = 2$ is chosen (besides that it is greater than 1, which makes intuitive sense), and the choice of $\beta$ in the experiments appears to be the result of picking the best $\beta$ after having tried many values and looking at the results. Without criteria for selecting these parameters, the new attack's strong performance could be difficult to reproduce/generalize in other settings.

4. *Possibility of "overfitting" to image data, neural networks, and/or particular data sets*. On a few occasions, there are signs that the proposed membership inference attack is somewhat tailored to classifying image data using neural networks. Although this task is popular, if not dominant, in the literature, the risk of "overfitting" can perhaps be made more explicit. For a few examples:
    * "simple transformations of x" at the end of Section 2.2 sounds straightforward for image data, but may not always make sense for other data types;
    *  while Sections 1 and 2 discuss membership inference attacks in very general terms, the empirical evidence is predominantly on classification of images by neural networks.
    * related to the second "weakness" above, the finiteness of "population" appears to be an artifact of the chosen data sets for empirical evaluation.

**Questions:**

The questions correspond to the "weaknesses" above.

1. Theoretical justification of the method.
   * For cancelling the $P(D|\theta')$ term in equation (3), while the intuition makes sense, can the argument be expressed in mathematical terms?
   * For $P(x)$, is the summation immediately below equation (4) an exact expression or an approximation? If it is an approximation, how would one sample $\theta'$ so that the sum is indeed a good approximation? Is any assumption needed for $P(\theta')$, say, discreteness or finiteness of support?

2. Dependence on reference records. While reference records are not always a direct input in other membership inference attacks, is there any way to assess whether the competing methods would depend more strongly or weakly on the number of reference records, compared to the new attack?

3. Lack of practical criteria for selecting input parameters $\gamma$ and $\beta$. Is $\gamma = 2$ always a good/acceptable choice? For a desired TPR/FPR value, is there a way to select $\beta$ without trying a range of values? If one has to try many values to make the decision, does it diminish the computation cost advantage over existing methods?

4. Possibility of "overfitting". Is there evidence, either theoretical or empirical, that the strong empirical performance observed in this paper can generalize to other models (particularly models other than neural networks)?

---

> ### Author Response · Authors · 2023-11-22
> **Response to Reviewer F7FX**
>
> Many thanks for the detailed comments.
>
> > 1. Theoretical justification
>
> We have revised Section 2. We substantially improved the organization of the section and provided interpretable presentation of our framework. First we present a template for the game and the MIA test which also applies to all major MIA methods in the literature.
>
> MIA is a hypothesis testing problem that assigns a **membership score** $\mathrm{Score}_\mathrm{MIA}(x; \theta)$ to every pair of $(x, \theta)$, and outputs a membership bit through comparing the score with a **threshold** $\beta$:
>
> $$\mathrm{MIA}(x; \theta) = 1 \text{ if } \mathrm{Score}_\mathrm{MIA}(x; \theta) \ge \beta$$
>
> For any given threshold $\beta$, the adversary's **power**, defined as the true positive rate of the attack, and **error** or the false positive rate, are quantified over numerous repetitions of this experiment.
>
> The ROC curves are computed by sweeping over all values of $\beta$.
>
> **Our work**
>
> We propose a novel method to compute $\mathrm{Score}_\mathrm{MIA}(x; \theta)$. The rest of the framework for the game and the test and their evaluation are the same as the prior work.
>
> **Revised Section 2**
>
> We have not made any change in the LR test, however, we made small changes to the way we arrive at the test which enables us to better explain our intuitions. The probability terms are also better explained. Here is the summary of the process:
>
> We compute the pairwise likelihood ratio as $\mathrm{LR}\_{\theta}(x, z) = \frac{\Pr(\theta | x)}{\Pr(\theta | z)}$. Our MIA score is $\mathrm{Score}\_\mathrm{MIA}(x; \theta) = \Pr_{z \sim \pi} \big( \mathrm{LR}_{\theta}(x, z) \ge \gamma \big)$.
>
> The term $\Pr(\theta | x )$ is the probability that the algorithm produces the model $\theta$ given that $x$ was in the training set, **while** the rest of the training set is randomly sampled from the population distribution $\pi$.
>
> Using the Bayes rule, we expand $\mathrm{LR}\_{\theta}(x, z)$ to $\left(\frac{\Pr(x | \theta)}{\Pr(x)}\right)\cdot\left(\frac{\Pr(z | \theta)}{\Pr(z)}\right)^{-1}$ with *no approximation*.
>
> The term $\Pr(x)$ is defined as the normalizing constant in the Bayes rule, and has to be computed by integrating the numerator of the Bayes rule over all models $\theta'$.
>
> $$\Pr(x) = \sum_{\theta'} \Pr(x | \theta') \Pr(\theta') = \sum_{D, \theta'} \Pr(x | \theta', D) \Pr(\theta' | D) \Pr(D) = \sum_{D, \theta'} \Pr(x | \theta') \Pr(\theta' | D) \Pr(D)$$
>
> We *empirically* compute $\Pr(x)$ as the empirical mean of $\Pr(x|\theta')$ by sampling *reference models* $\theta'$, each trained on random datasets $D$ drawn from the population distribution $\pi$.
>
> We hope the process is more clear. Compared to the previous version, we incorporate the effect of other data points in the training set in the computation of $\Pr(\theta | x )$. Note that, the attack process and our final test remained unchanged.
>
> > 2. Reference records
>
> **Clarification**: The percentage refers to the size of the reference models' training set rather than the entire population. So, a portion of the samples used for training reference models is selected as the set of reference records. We have revised the relevant section (Appendix A.8) to accurately reflect this.
>
> **Dependence on number of records**: All attacks assume they have a large number of reference points, which (except the Home et al and Attack-P in Ye etal 2022 methods) they use all the reference records to train reference models. In our attack, we show that a fraction of these data points should also be used as $z$ reference points in our attack to boost the relativity of the likelihood of observing $\theta$ under $b=0$ versus $b=1$.
>
> > 3. Selecting $\gamma$ and $\beta$
>
> To compute ROC curve, all methods sweep over values of $\beta$. In our method (as opposed to e.g., Carlini etal 2022), the exact value of each $\beta$ is interpretable and is calibrated: As shown in Figure 14, for $\gamma=1$, the value of $\beta$ is $1-FPR$.
>
> We added Figure 12 (Section A.9) to show the stability regarding $\gamma$. Our attack's performance remains consistent across different values of $\gamma$.
>
> Interpretation of $\gamma$ is that by increasing it, we reduce the attack FP for the same $\beta$. But, if we are sweeping over $\beta$, the ROC curve does not change significantly.
>
> > 4. Overfitting
>
> We considered diverse benchmark datasets, and also used the Purchase-100, which is a non-image dataset in MIA literature.
>
> To address your comment, we explored the performance of attacks across various ML algorithms. In new Appendix A.6, we analyzed the Gradient Boosting Decision Tree algorithm on Purchase-100.
>
> In new Appendix A.3 and A.4, we analyze the effect of data distribution and architecture shifts (using a different dataset/architecture for reference models than the target model). In all cases, our method consistently outperforms other attacks, and the gap of performance even increases, i.e., other attacks are more susceptible to these changes.

---

### Official Review · Reviewer_qcX6 · 2023-10-31

**Soundness:** 2 fair
**Presentation:** 2 fair
**Contribution:** 2 fair
**Rating:** 5
**Confidence:** 3

**Summary:**

This paper studies the problem of membership inference attack (MIA). More specifically, the authors propose a new MIA method, RMIA, which can achieve better TPR-FPR tradeoffs comparing previous methods. Empirical results validate the effectiveness of the proposed method.

**Strengths:**

The strength of the paper is as follows:
1. The authors propose a new MIA method, which is based on a new approximation of the likelihood ratio (LR).
2. The computation of the proposed LR seems to be easy to implement.
3. The empirical results across different datasets validate the advantages of the proposed method.

**Weaknesses:**

The weakness of the current paper:
1. The presentation of the paper need to be improved. For example, the comparisons between the proposed method and the previous methods needs to be further clarified, especially for the method proposed by Carlini et at., 2022.
2. It is unclear why the authors can assume that $Pr(D|\theta^\prime)$ to be a constant.
3. The authors do not test the methods when the model is differentially private.

**Questions:**

I find the idea of the paper is interesting and the results seems to be promising. However, I have the following additional questions regarding the current paper:
1. In Definition 1, you assume a fair coin $b$. What if the probability of the data being a member or not is not $0.5$, and whether your method can be applied to this case?
2. For the parameters $\beta$ and $\gamma$, how sensitive of these parameters and whether it is hard to find the optimal parameters?
3. What are the standard deviations of your report results? Do you have some confidence intervals in your plots?
4. For the predication probability functions ($Pr(x|\theta)$), how sensitive are those hyperparameters?
5. When you compute $Pr(x)$ for the offline method, what is the computational cost and how the results will be affect by the linear approximations?
6. If you continue to increase the reference models, how will your methods look like compared to the method proposed by Carlini et at., 2022?

---

> ### Author Response · Authors · 2023-11-22
> **Response to Reviewer qcX6**
>
> Many thanks for detailed questions and comments.
>
> ## Weaknesses
>
> > 1. The presentation of the paper need to be improved. For example, the comparisons between the proposed method and the previous methods needs to be further clarified, especially for the method proposed by Carlini et at., 2022.
>
> We have revised Section 2. The updated version should address this comment. We show that all major MIA methods can be presented within the same structure, as we present in the beginning of Section 2. The major difference between attacks is the way they compute their MIA score. Please see updated Section 2.3 and the newly added Table 1. As we describe in the text, the LR corresponding LiRA has an averaging factor over all $z$, while in our attack we split LR for all $z$, which gives us a better distinguishability of $x$ and non-members. Also, our Bayes computation of LR provides a significant edge over direct computations of LR, as shown in newly added Figures 20 and 21. The new plots in Figure 6 also show the significant power of our attack compared to other attacks.
>
> > 2. It is unclear why the authors can assume that $\Pr(D | \theta')$ to be a constant.
>
> Appendix A.13 and Figure 22 show that the variance of this quantity is extremely small across different $\theta'$. So, their changes could be ignored without hurting our test power. Also, in the revised explanation of our method, we provide a different yet equivalent and more interpretable derivation of $\Pr(x)$ and $\Pr(z)$, where the effect of $D$ is more intuitive. Note that our computations of LR have not changed.
>
> > 3. The authors do not test the methods when the model is differentially private.
>
> We have included a new section, Appendix A.5, to assess the impact of using DP-SGD on the performance of attacks with various combinations of DP-SGD’s noise multiplier, clipping norm, and privacy budgets. As shown in Figure 9, our attack consistently outperforms others, especially in more stringent settings.
>
> ## Questions
>
> > 1. In Definition 1, you assume a fair coin $b$. What if the probability of the data being a member or not is not $0.5$, and whether your method can be applied to this case?
>
> This is the standard approach for all MIAs. Changing $b$ doesn't change the attacks, but only the interpretation of the TPR-FPR curve.
>
> Something interesting, however, is that if we are concerned with the fact that in practice there are orders of magnitude more non-members than members for any model, we need to pay attention to 2 factors in any MIA: The cost of performing attack on any new data point, and their TPR versus FPR (especially for low FPR and high TPR regions).
>
> > 2. For the parameters $\beta$ and $\gamma$, how sensitive of these parameters and whether it is hard to find the optimal parameters?
>
> As discussed in the revised Section 2, $\beta$ is common among all attacks, and will be used to generate the ROC curve.
>
> We added Figure 12 (Section A.9) to show the stability regarding $\gamma$. Our attack's performance remains consistent across different values of $\gamma$.
>
> > 3. What are the standard deviations of your report results? Do you have some confidence intervals in your plots?
>
> Added for Tables 2, 3, 4.
>
> > 4. For the predication probability functions ($\Pr(x | \theta)$), how sensitive are those hyperparameters?
>
> We conducted a performance sensitivity analysis of our attack with respect to different hyperparameters of computing probabilities. Please see Figure 15. Firstly, as indicated in Table 6 of Section A.10.2, using different functions has a negligible impact on the result of our attack. Furthermore, as illustrated in the figure, our attack's performance remains robust against variations in the soft-margin (m), temperature (T), and higher orders (n).
>
> > 5. When you compute $\Pr(x)$ for the offline method, what is the computational cost and how the results will be affect by the linear approximations?
>
> Please see new explanations in Appendix A.10.3. We compute $\Pr(x)$ only using the available reference models. If we have 1 or 2 reference models, we use fine-tuning to compute the gap between $\Pr(x)\_{OUT}$ and $\Pr(x)\_{IN}$. Also the new Figure 16 shows that the attack is not very sensitive to small changes to $a$. Notably, overestimating the gap between $\Pr(x)\_{OUT}$ and $\Pr(x)\_{IN}$ has minimal impact.
>
> > 6. If you continue to increase the reference models, how will your methods look like compared to the method proposed by Carlini et at., 2022?
>
> Please see the new Figure 6. In the offline setting, LiRA Carlini 2022 cannot reach the performance of RMIA and other attacks (e.g., Ye 2022). In the online setting, with a large number of reference models it approaches RMIA. Note that even in that setting, RMIA offline is almost as good as LiRA online (with significantly higher cost of training hundreds of reference models per *new* MIA query). Please also see Figures 2, 3, 4, 5 for the ROC curves, as we increase the number of reference models.

---

### Official Review · Reviewer_Ju68 · 2023-11-01

**Soundness:** 3 good
**Presentation:** 3 good
**Contribution:** 3 good
**Rating:** 8
**Confidence:** 4

**Summary:**

This article suggests an improvement to Membership Inference Attacks (MIAs) while ensuring low costs. The authors consider the adversary-challenger model wherein given two samples (target and random), the adversary attempts to find whether a model is trained on the target sample or the random one. By designing a simplified, low-compute likelihood function by allowing access to reference models, the adversary in this paper can identify the presence of the target sample (in the given model's train set) with a higher probability compared to previous SOTA work. Empirical results verify the usefulness of this work. As with all MIAs, the impact of this work is in designing superior ways of testing the privacy claims of a method.

**Strengths:**

1. The core usefulness of the method is based on the simplification of the likelihood function (by leveraging reference models). The likelihood exhibits superior qualities and requires lower computations compared to previous methods.
2. Unlike previous methods, the new method does not require the assumption that the target and random sample have the same predictive probability for any model. Essentially the new method captures that the random and target samples can be quite different. Although this point is highlighted in section 2.3 (para 3) it will be nice to see some basic experiments on how much different two samples can be. For example, consider showing the predictive power differences between the boundary points nearer to the decision boundary v/s interior points.
3. The suggested adversary can be developed for any given training algorithm/model. More importantly, the method is quite straightforward (except for access to reference models, see weaknesses section).
4. The method in this work achieves higher True Positive Rates (TPRs) across all False Positive Rates (FPRs) compared to previous methods.
5. The overall strong empirical results verifies the method's strengths.

**Weaknesses:**

1. It is unclear whether the approximation of the likelihood always holds. The idea is that different reference models exhibit similar predictive distributions. Is it assumed that the reference models are trained well and over a large sample size for this assumption to hold? Essentially, having some clarity about the assumptions of the reference models will be useful for judging the adversary's capacity.
2. Continuing point 1, how hard is it for the adversary to access/train such reference models and is it a standard assumption in literature? A brief discussion about the adversary's strength will be helpful.
3. The authors mention that they incur lower costs as they do not have to train models including the target sample. However, does the training of reference models not incur additional costs? Providing a simple cost comparison discussion will help.

**Questions:**

My main question is regarding the inclusion of reference models as highlighted in the weaknesses section. Answering the questions (in the weaknesses section) will alleviate most of my concerns. Otherwise, the paper is well-written and provides a straightforward method for designing better, low-cost MIAs.

---

> ### Author Response · Authors · 2023-11-22
> **Response to Reviewer Ju68**
>
> Many thanks for the questions. We have further improved the clarity of our method in Section 2. The method remains exactly the same, however, we first clearly explain what the structure of the attack and its evaluation is, which is common among all the major MIA methods. Then, we compute our LR which helps computing the MIA score.
>
> > 1. It is unclear whether the approximation of the likelihood always holds. The idea is that different reference models exhibit similar predictive distributions. Is it assumed that the reference models are trained well and over a large sample size for this assumption to hold? Essentially, having some clarity about the assumptions of the reference models will be useful ...
> > 2. Continuing point 1, how hard is it for the adversary to access/train such reference models and is it a standard assumption in literature? A brief discussion about the adversary's strength will be helpful.
>
> **Why reference models?** Using reference models (also referred to as shadow models in the literature) is the cornerstone of the SOTA membership inference attacks. Effectively, without reference models, it is very hard to quantify how much inclusion of a given data point $x$ in the training set of a model could have influenced the model. Reference models act as approximations for what the model could have been with or without $x$ in the training set. Without the use of reference models, the MIA attacks would be only able to capture the leakage due to the average generalization gap in the models. With the use of reference models, the MIA attacks are able to capture the leakage due to the memorization of a specific target data point, thus their strength. However, some MIA methods (notably LiRA Carlini et al) require a very large number of models to be trained for each target model, rendering them less useful in practice.
>
> **Performance variation of our reference models.** We show these results in Appendix A.13. As it is shown, the average probability of population data has a very small variance across different reference models.
>
> **Attack performance under low quality reference models.** Your comment raises a good point regarding the stability of tests with respect to quality of reference models. Thus, we added new experiments to test this. In Appendix A.3, we demonstrated that training reference models with data distribution shifts (namely, using a completely different dataset than the target model) still yields significantly better results for our proposed attack compared to other attacks. Additionally, in Appendix A.4 (Figure 8), we showed that our attack outperforms other attacks even when reference models are trained with different architectures than the target model. Interestingly, the performance gap between our attack and others increases in such cases. Another important point is that we deliberately avoided taking advantage of overfitted models, as the train-test accuracy gap in our trained model is smaller compared to experiments in prior works. It's worth noting that the performance of attacks naturally increases with overfitted models. As reported in Section 3.1, we trained several models with varying test accuracies, from 67% to 92%, to show the superiority of our attack.
>
> **Performance under very few reference models.** We demonstrated that even in the extreme case of using only 1 offline reference model, our attack can perform well and completely overcome other attacks, for example achieving at least 28% better AUC and 2x-4x higher TPR at zero FPR across all datasets, compared to the well-known LiRA attack.
> A comparison of the properties of the MIA test scores assigned by different attacks (as presented in Figure 17-19 in Appendix A.11) highlights that our attack achieves superior separation between scores of members and non-members, particularly when limited to a few reference models.
>
> > 3. The authors mention that they incur lower costs as they do not have to train models including the target sample. However, does the training of reference models not incur additional costs? Providing a simple cost comparison discussion will help.
>
> Cost of reference models needs to be evaluated along with their added value for the attack outcome. Please see the newly added Figure 6, where we show the AUC of attacks, when we increase the number of reference models.
>
> In the online setting, adversary needs to train $k$ additional models **per new MIA query**. This is a huge cost. Carlini et al LiRA attack performs poorly for a small to moderate number of reference models, and can achieve a comparable attack performance to ours only when the number of models is significantly large.
>
> In the offline setting, adversary trains $k$ reference models in advance, prior to knowing the MIA queries. Our attack clearly outperforms other attacks, and with a few reference models it performs very close to its best results.
>
> Using a couple of reference models (offline) RMIA outperforms the prior attacks (online or offline)

---

### Official Review · Reviewer_HGWk · 2023-11-01

**Soundness:** 2 fair
**Presentation:** 1 poor
**Contribution:** 2 fair
**Rating:** 5
**Confidence:** 3

**Summary:**

This paper suggests a new likelihood ratio loss-based membership inference attack. The authors suggest that their attack differs from SOTA loss based attacks that also rely on the likelihood ratio by additionally incorporating a variety of reference points as opposed to just one reference point in the standard LiRA attack (e.g., Carlini et al (2021)). The authors’ suggested attack clearly outperforms the SOTA attacks by a large margin on standard benchmark datasets like CIFAR10, CIFAR100, Purchase100 and CINIC10.

Despite the test’s strong empirical performance, I am hesitant to provide a more favourable evaluation of the proposed method at this point. This is since there are insufficient details to properly understand how the test is conducted in practice. In particular, the paper does neither provide pseudo code for their attack nor does it describe the step-by-step computation of the test statistic.  If authors could provide clarifications, I may be willing to revise my evaluation.

**Strengths:**

**New loss-based attack**: The authors propose a new attack that uses a model’s losses and that seems to outperform SOTA attacks. The attack uses a variety of reference points to calibrate the distinguishability between x and any z when conditioned on $\theta$.

**Comprehensive empirical evaluation**: The demonstrated empirical evaluation effectively compare the proposed test’s performance against other state-of-the-art attacks that are based on the loss. The results are shown across standard benchmark data sets including CIFAR10, CIFAR 100, Purchase100 and CINICO10 on which the proposed attack seems to outperform SOTA by a large margin.

**Weaknesses:**

**Missing details**: The paper does not provide pseudo code for their suggested attack. Neither is the exact computation of the test statistic described. This makes it difficult to fully appreciate the work’s results. Providing further details on this would help to follow the author’s argument more easily. Further, the discussion on the mechanism of the author’s proposed attack in section 2.3 is neither supported by empirical evidence nor accompanied by a theoretical analysis, that would link the discussed probabilities to the power of the likelihood ratio test, and is thus difficult to follow.

**Comparison**: The MIA attack setup described in this work is different from the LiRA attack described in previous work (e.g., Carlini et al (2021)). In the LiRA attack, the attacker does not have the capacity to poison the dataset. Hence, since the attacks run under different threat models the comparison may be misleading.

**Questions:**

- How is the indistinguishability game that you propose required for your attack? Why do you require that your game be different from the standard MI attack game proposed by Yeom et al (2018), used in Carlini et al (2021) and recently analysed by Leemann et al (2023)?
- Aren’t p(z) and p(x) just the prior probabilities of observing x and z, respectively?

----
**Additional references**

Leemann et al (2023), „Gaussian Membership Inference Privacy”, 37th Conference on Neural Information Processing Systems (NeurIPS)

---

> ### Author Response · Authors · 2023-11-22
> **Response to Reviewer HGWk**
>
> Thanks for the comments and questions.
>
> ## Details of the Game and the Test
> > Missing details: The paper does not provide pseudo code for their suggested attack. Neither is the exact computation of the test statistic described.
> - We revised Section 2 thoroughly to better explain the game and different steps of the attack.
> - We added Appendix A.1 which includes the pseudo-codes and detailed step-by-step computations needed for the attack.
>
> ## Membership inference game
> > The MIA attack setup described in this work is different from the ... previous work ... that does not have the capacity to poison the dataset.
>
> > How is the indistinguishability game that you propose required for your attack? Why do you require that your game be different ...?
>
> We have revised the text to clarify this (which we admit was confusing). Our game and the test experiments are exactly the same as that of the prior work, and we do not assume any poisoning. Please see Section 2, where in the beginning of the section we describe the foundation of the test.
>
> **Membership Inference Game**
>
> Let~$\pi$ be the data distribution, and let $\mathcal{T}$ be the training algorithm.
> - The challenger samples a training dataset $S \sim \pi$, and trains a model $\theta \sim \mathcal{T}(S)$.
> - The challenger flips a fair coin $b$. If $b = 1$, it randomly samples a data point $x$ from $S$. Otherwise, it samples $x \sim \pi$, such that $x \notin S$. The challenger sends the target model $\theta$ and the target data point $x$ to the adversary.
> - The adversary, having access to the distribution over the population data $\pi$, outputs a membership prediction bit $\hat{b} \leftarrow \mathrm{MIA}(x; \theta)$.
>
> **Membership Inference Attack**
>
> A membership inference attack is a hypothesis testing problem that assigns a **membership score** $\mathrm{Score}_\mathrm{MIA}(x; \theta)$ to every pair of $(x, \theta)$, and outputs a membership bit through comparing the score with a **threshold** $\beta$:
>
> $$\mathrm{MIA}(x; \theta) = 1 \text{ if } \mathrm{Score}_\mathrm{MIA}(x; \theta) \ge \beta$$
>
> For any given threshold $\beta$, the adversary's **power**, defined as the true positive rate of the attack, and **error** or the false positive rate, are quantified over numerous repetitions of this experiment.
>
> **Our work**
>
> We propose a novel method to compute $\mathrm{Score}_\mathrm{MIA}(x; \theta)$. The rest of the framework for the game and the test and their evaluation are the same as the prior work.
>
> ## Probabilities used in the likelihood ratio
> >Aren’t p(z) and p(x) just the prior probabilities of observing x and z, respectively?
>
> We have revised the text. $\Pr(x)$ is not the same as $\pi(x)$, which is rather the prior distribution over $x$. The term $\Pr(x)$ is defined as the normalizing constant in the Bayes rule, and has to be computed by integrating the numerator of the Bayes rule over all models $\theta'$.
>
> $$\Pr(x) = \sum_{\theta'} \Pr(x | \theta') \Pr(\theta') = \sum_{D, \theta'} \Pr(x | \theta', D) \Pr(\theta' | D) \Pr(D) = \sum_{D, \theta'} \Pr(x | \theta') \Pr(\theta' | D) \Pr(D)$$
>
> We compute $\Pr(x)$ as the empirical mean of $\Pr(x|\theta')$ by sampling *reference models* $\theta'$, each trained on random datasets $D$ drawn from the population distribution $\pi$.
>
> Note in the equation above $\Pr(x | \theta', D)$ is equal to $\Pr(x | \theta')$ because *given* a model $\theta'$, (the prediction on) a data point is independent of the training set $D$ (i.e., conditional independence between $x$ and $D$ given $\theta'$)
>
> ## Discussion on the mechanism
> > The discussion on the mechanism of the author’s proposed attack in section 2.3 ...
>
> We have revised Section 2.3 and added new analysis and discussions. The discussion is around how other methods can be simply constructed from ours via averaging or simplifications (So, we are not deriving any new theoretical results there). Here is our clarification.
>
> MIA is a hypothesis test problem. In **our method**, we consider the worlds associated with null hypothesis to be the worlds in which $x$ is not in the training set and *instead* a different data point $z$ is included. This allows us to model the null hypothesis (non-member) worlds in a more fine-grained way, as opposed to the prior SOTA.
>
> In Section 2.3, we also provide a direct way to compute our LR which makes our attack more easily comparable to LiRA.
>
> Our pairwise LR: $$\frac{\Pr(f_\theta(x), f_\theta(z) | x)}{\Pr(f_\theta(x), f_\theta(z) | z)}$$
>
> LiRA Carlini et al LR: $$\frac{\Pr(f_\theta(x) | x)}{\Pr(f_\theta(x) | \bar{x})}$$
>
> $\bar{x}$ means the $x$ is *not* in training set.
>
> We observe that the denominator in LiRA LR is the average case for our LR averaged over all $z$. This reduces the power of LiRA's test. Also, as we show in Appendix A.12, a direct computation of LR requires a large number of reference models. The combination of a pairwise LR and its computation using the Bayesian approach results in higher power and lower cost in our attack.

---

### Author Response · Authors · 2023-11-22
**Changes in the revised version**

Many thanks for all the questions and comments.

We have revised the paper and added new discussions, and many new experimental results in response to reviewers’ comments.

Here is the short list of changes in the revision:

1) Improving the organization of Section 2, and improving the explanation of our attack and its intuitions,
2) New Appendix A.1 to show pseudocode and details of test statistic computation,
3) Results of using different NN architectures (illustrated in Figure 7-8, Appendix A.4),
4) Results of attacking DP models (shown in Figure 9, Appendix A.5),
5) Results of training with ML algorithms different from NN (depicted in Figure 10, Appendix A.6),
6) Results of evaluating with interior (easy and typical) and boundary (hard and atypical) target samples (depicted in Figure 11, Appendix A.7),
7) Performance sensitivity of our attack with respect to hyper-parameters (shown in Figure 12, 15, and 16),
8) An alternative direct way to compute our likelihood ratio, in Section 2.3, and its empirical results (discussed in Appendix A.12).

---

### Meta-Review · Area_Chair_mtdC · 2023-12-12

**Metareview:**

This paper introduces a membership inference attack game featuring a novel attack (RMIA) that effectively utilizes both reference models and population data in its likelihood ratio test. The algorithm demonstrates superior test power, particularly at extremely low false-positive error rates, outperforming prior methods across the entire TPR-FPR tradeoff curve.

While the reviewers pointed out several limitations of the paper, the authors' rebuttal addressed many of the concerns which is greatly appreciated. However, the paper's presentation still remains a concern, particularly due its empirical nature whose primary contribution lies in proposing a new membership inference algorithm. The reproducibility of the proposed algorithms and clarity of presentations are very important in such works; however, the current presentation of the main paper impedes a re-implementation. Crucial details, such as the pseudo-code and guidelines for choosing \gamma, are deferred to the appendix, along with other essential assumptions (e.g., equation 10 in the appendix), all of which hinder a comprehensive understanding of the paper and will pose challenges for readers to derive meaningful insights.

We encourage the authors to resubmit the work after addressing these concerns and also incorporating the remaining suggestions of the reviewers.

**Justification For Why Not Higher Score:**

N/A

**Justification For Why Not Lower Score:**

N/A

---

### Decision · Program_Chairs · 2024-01-16

Reject